# The protein interactome of the citrus Huanglongbing pathogen *Candidatus* Liberibacter asiaticus

Erica W. Carter[1,2], Orlene Guerra Peraza[1] & Nian Wang [1,3] ✉

The bacterium *Candidatus* Liberibacter asiaticus (CLas) causes citrus Huanglongbing disease. Our understanding of the pathogenicity and biology of this microorganism remains limited because CLas has not yet been cultivated in artificial media. Its genome is relatively small and encodes approximately 1136 proteins, of which 415 have unknown functions. Here, we use a high-throughput yeast-two-hybrid (Y2H) screen to identify interactions between CLas proteins, thus providing insights into their potential functions. We identify 4245 interactions between 542 proteins, after screening 916 bait and 936 prey proteins. The false positive rate of the Y2H assay is estimated to be 2.9%. Pull-down assays for nine protein-protein interactions (PPIs) likely involved in flagellar function support the robustness of the Y2H results. The average number of PPIs per node in the CLas interactome is 15.6, which is higher than the numbers previously reported for interactomes of free-living bacteria, suggesting that CLas genome reduction has been accompanied by increased protein multi-functionality. We propose potential functions for 171 uncharacterized proteins, based on the PPI results, guilt-by-association analyses, and comparison with data from other bacterial species. We identify 40 hub-node proteins, including quinone oxidoreductase and LysR, which are known to protect other bacteria against oxidative stress and might be important for CLas survival in the phloem. We expect our PPI database to facilitate research on CLas biology and pathogenicity mechanisms.

Citrus Huanglongbing (HLB) is one of the most destructive plant diseases worldwide. HLB has devastated the citrus industry in many regions, including Florida, US, and no effective and economic disease control measures are available for these HLB-endemic citrus production regions[1,2]. HLB is caused by *Candidatus* Liberibacter asiaticus (CLas), *Ca*. L. americanus, and *Ca*. L. africanus, with CLas being the most prevalent[3]. *Ca*. Liberibacter species are endophytic bacteria that colonize the phloem tissues of citrus plants and are transmitted by the Asian citrus psyllid *Diaphorina citri*. The environments inside the phloem tissue and psyllids are nutrient-rich. Consequently,

*Ca*. Liberibacter species have undergone reductive evolution and have considerably smaller genomes (approximately 1.2 MB) than their culturable, free-living relatives, such as *Agrobacterium*[4,5]. Cultivation of HLB pathogens in artificial media has not been achieved despite extensive effort[6], which prevents its genetic manipulation and impedes the understanding of their biology and pathogenicity mechanism.

The CLas str. Psy62 (NC_012985.3) genome encodes approximately 1136 genes, 1027 of which are non-redundant proteins. Sixty percent (612 ORFs) of these coding regions have predicted functions based on sequence similarities with characterized proteins from

[1]Citrus Research and Education Center, Institute of Food and Agricultural Sciences, University of Florida, Lake Alfred, FL, USA. [2]Department of Plant Pathology, Institute of Food and Agricultural Sciences, University of Florida, Lake Alfred, FL, USA. [3]Department of Microbiology and Cell Science, Institute of Food and Agricultural Sciences, University of Florida, Lake Alfred, FL, US. ✉e-mail: nianwang@ufl.edu

culturable microorganisms. However, 415 CLas proteins have no known functions; these were defined by Duan et al. as 69 with general function only, 46 had no known function, and 300 without a COG family[5]. Investigating protein functions, particularly those with unknown roles, is critical for elucidating the biology and pathogenicity of CLas and developing effective strategies to combat this notorious citrus disease. Protein–protein interactions (PPIs) are fundamental to all cellular processes and machinery. PPIs modify enzyme and protein activities, catalyze metabolic reactions, and activate signaling pathways[7]. Exploring PPIs has also been used to investigate biological pathways, especially interactions between proteins of known and unknown function. These PPIs provide informative clues for pathway and function prediction[8,9]. Furthermore, targeting PPIs can be used to develop novel antimicrobials[10]. The Yeast Two-hybrid system (Y2H) is the most commonly used approach to investigate interactomes and has been elegantly applied to various organisms, such as animals including *Drosophila melanogaster*[11] and *Caenorhabditis elegans*[12,13]; plants including *Arabidopsis thaliana*[14]; fungi including *Saccharomyces cerevisiae*[8]; viruses including potyviruses Soybean mosaic virus[15]; and bacteria including *Streptococcus pneumoniae*[16], *Treponema pallidum*[17], *Campylobacter jejuni*[18], *Synechocystis sp.*[19], *Mycobacterium tuberculosis*[20], *Mesorhizobium loti*[21], *Escherichia coli*[22], *Bacillus subtilis*[23], and *Helicobacter pylori*[24]. Affinity Purification Mass Spectrometry (AP-MS) has also been used to study the *Mycoplasma pneumoniae* interactome[7,25]. However, AP-MS detects PPIs in the native organism, revealing protein complex topology and not necessarily binary topology, whereas Y2H bypasses this limitation by identifying the binary PPIs, which has been used to elucidate the binary PPIs found in AP-MS studies[22]. Interactome studies (Supplementary Table 1) have led to significant progress in inferring the function of unknown proteins, elucidating protein organization of a cell, identifying novel functions of known proteins, and, in some cases, are used to find

interaction points between pathogenic organisms and their eukaryotic hosts[26–32].

Here, we conducted a genome-wide Y2H interactome study for CLas, identifying over 4000 PPIs and proposing potential functions for 171 uncharacterized proteins.

## Results

### High-throughput Y2H screening

We conducted Y2H assays to identify the CLas interactome. We used full-length open reading frames (ORFs) for yeast vector construction to increase the possibility of identifying biologically relevant interactions. We successfully amplified 974 CLas ORFs (Supplementary Data 1) and cloned 942 ORFs into Y2H expression vectors (Supplementary Fig. 1A). In total, 926 "bait" proteins in pGEO_BD, which was modified from pGBKT7_BD (Supplementary Fig. 1B), and 950 "prey" proteins in pGAD_T7. After the removal of autoactivating constructs (Supplementary Fig. 2), 916 "bait" and 936 "prey" constructs remained suitable and utilized in the screening, representing 91% of genes encoded by CLas. We constructed the CLas binary interactome using a three-phase array screening method (Fig. 1A). The first two pooled array phases removed non-interactors. Proteins interacting on the most stringent media, QX (-trp, -leu, -ade, -his, +XαGal) and QXA (-trp, -leu, -ade, -his, +XαGal, +AbA) in the first two screening phases were used in the pairwise array in the third screening phase. The final (third phase) screening between 617 baits and 466 preys resulted in 4245 interactions between 542 proteins (Supplementary Data 2), covering 52.8% of CLas genes, denoted as the CLas_whole network (Fig. 1B). Among the 542 proteins, 371 had known functions, whereas 171 were uncharacterized proteins (Supplementary Data 3; Tables 1 and 2). The pairwise screening enabled us to identify all potential interactions from the mini-pooled arrays and detect the interacting partners. In addition, pairwise rescreening eliminated probable promiscuous

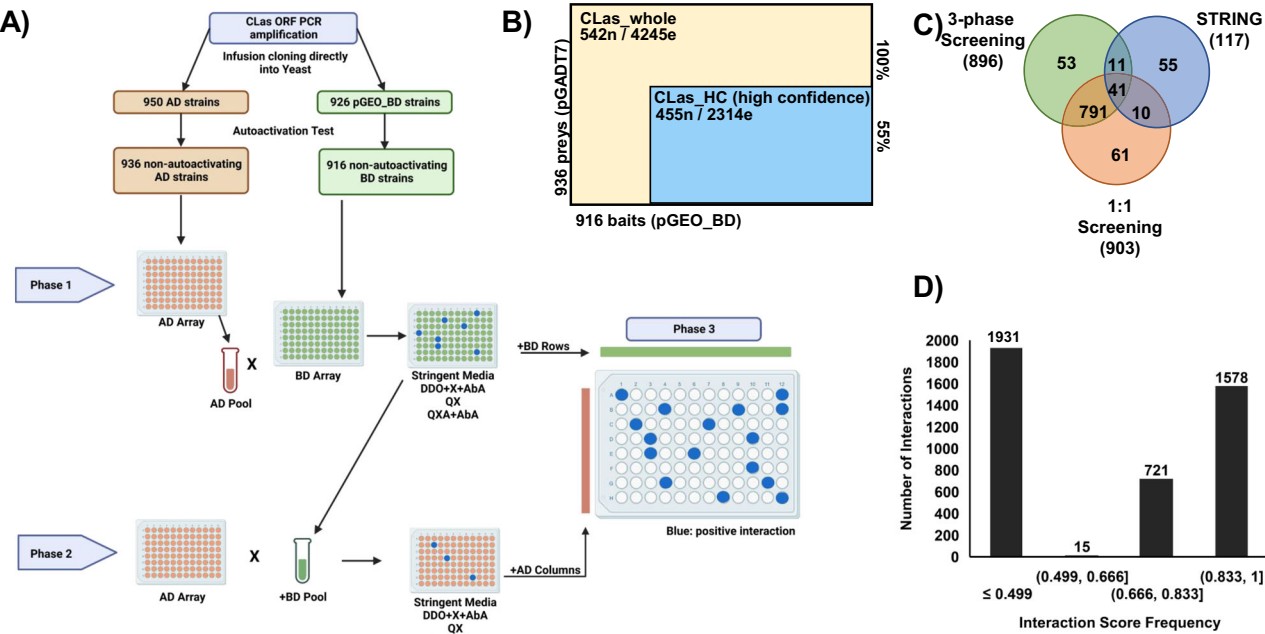

**Fig. 1 | The CLas Y2H interactome screening, validation, and confidence scores. A** The three-phase 96-well plate screening pipeline for constructing the CLas_whole network. Phase one and two systematically screened pools and individual yeast constructs to identify potential interacting protein partners. All interacting partners were identified in the third phase in a pairwise screening; phase 1 and 2 interacting proteins were screened pairwise against all other phase 1 and 2 interacting proteins to produce the binary data. This figure was generated using BioRender. **B** CLas Y2H three-phase screening results. n node, e edge, 100% indicates:

CLas_whole network; 55% indicates: CLas_HC (high confidence) network portion of the CLas_whole network. **C** Venn diagram showing the overlap of interactions detected in the three-phase screening, an independent pairwise screening, and predicted interactions for 163 CLas proteins; predicted interactions were downloaded from the String consortium. **D** Interaction confidence score frequency across the CLas_whole network. Interactions with a confidence score ≥0.5 are considered high confidence; denoted as the CLas_HC network.

proteins from the dataset by observing PPIs resulting from promiscuous proteins.

## CLas interactome coverage and confidence scores

The robustness of the Y2H datasets is imperative for utilizing the PPI information[33–35]. We reduced false positives by pairwise rescreening and stringent selection of the CLas PPI network in the three-phase screening. To evaluate the sensitivity and reproducibility of the three-phase method, we subjected a random set of 326 yeast constructs (163 proteins in each vector) to an independent pairwise Y2H screening. We observed 832 reproducible interactions between the three-phase and pairwise screening, representing 92% of the interactions detected between those 163 proteins in the CLas_whole network.

We further employed known PPIs in the STRING-db to evaluate the sensitivity of the two screening assays. Limited information is available for CLas in the STRING database, with no experimentally determined interactions reported for CLas specifically. However, there were 474 and 369 experimentally determined interactions between homologous, interolog proteins (iPPIs) and interacting protein family interologs (iCOGs) at confidence cutoffs ≥0.4 and ≥0.9, respectively (Table 1; Supplementary Table 2)[36,37]. Our results showed that both Y2H screening methods had comparative sensitivity in detecting predicted CLas interactions based on known PPIs of homolog proteins. The three-phase screening detected 52 whereas the pairwise approach identified 51 of the 117 predicted interactions among these 163 proteins with 41 detected by both methods (Fig. 1C, Supplementary Data 4).

Using the interactions detected in the two Y2H screening methods, we calculated the false positive rate for the high-throughput three-phase Y2H screening as 2.9% and the false negative rate as 55.6%. Thus, our three-phase Y2H is stringent against false positives in identifying putative PPIs of CLas. However, though the high throughput three-phase screening is efficient, it also brings a significant false negative rate of 55.6%. Because of the highly reproducible PPIs (92%) between the three-phase and pairwise screenings (Fig. 1C), it is probable that the false negative rate is an intrinsic limitation of the Y2H assay as previously reported for other binary PPI assays[14,38,39]. Nevertheless, the low false positive rate rendered our three-phase Y2H data highly robust to analyze the CLas interactome and infer the protein function of hypothetical proteins, further supported by the CLas interactome specificity of 97.1%, where very few non-interactors were detected.

To evaluate the robustness of individual PPIs, we calculated interaction confidence scores for the CLas_whole network PPIs using iPPIs as the positive reference set. The interologs were obtained by querying the STRING-db for experimentally determined PPIs from *Agrobacterium radiobacter, B. subtilis, C. jejuni, E. coli, H. pylori, L. crescens, M. genitalium, M. loti, M. pneumoniae, S. cerevisiae, S. meliloti, T. pallidum*, and iCOGs. There were 635 iPPIs between 208 protein pairs in the CLas_whole network (Supplementary Data 5 and 6). We used 100 PPIs corresponding to interologs with the highest colony scores to represent the positive reference set for logistic regression analysis and assign interaction confidence scores to the CLas_whole PPIs. In total, 2314 interactions between 455 proteins demonstrated high confidence scores (≥0.5, CLAS_HC), representing 55% of the total interactions detected in the CLas_whole network and 44% of the CLas ORFs (Fig. 1B, D, Supplementary Fig. 3, Supplementary Data 7).

To further verify our Y2H PPIs, we conducted pull-down assays for nine PPIs focusing on flagellar proteins because of the crucial roles of CLas motility in the HLB pathosystem[40]. Consistent with the Y2H result, nine tested PPIs were confirmed (Fig. 2), supporting the high quality of the three-phase Y2H screening.

## Network topology

Protein network topology provides valuable information for protein functions[41,42]. We visualized and analyzed the topology of the CLas Y2H interactome (Fig. 3A–C)[43,44]. The degree distribution topology of CLas_whole had a scale-free organization, typical of a scale-free PPI network (Fig. 3D)[28]. The average node connectivity was 15.664 (average node degree of 7.8) in the CLas_whole network; CLas_HC has an average node connectivity of 10.171 (average node degree of 5.1). The CLas network had many node neighbors, consistent with the notion that protein function complexity increases with genome reduction (Fig. 3F)[7,45]. The CLas_HC network maintained a degree distribution and topological coefficient distribution like that of the CLas_whole network (Fig. 3D, G). However, the clustering coefficient and betweenness centrality distribution of CLas_HC were less dense than the CLas_whole with fewer crosstalk between nodes (Fig. 3E, F)[6].

Highly connected nodes (hubs) are more likely to be responsible for maintaining the overall connectivity of the network. They are more likely to be essential genes or involved in critical PPIs than non-hub nodes[46]. We identified 40 hub nodes in the CLas_whole network. Using betweenness centrality and degree distribution, we manually curated 27 hub nodes (Fig. 3D, E)[46]. We validated the hubs using the Maximal Clique Centrality (MCC) and MCODE algorithms[47,48]. MCODE clustering confirmed 26 of our 27 manually curated nodes as hubs with scores ≥4. The MCC clustering algorithm confirmed 14 hub nodes defined by manual curation and MCODE clustering and suggested 13 additional hub proteins (Fig. 3B, Supplementary Data 8). Interestingly, eight of the hub proteins were reported to be essential for *L. crescens* [CLIBASIA_RS05300: terminase; CLIBASIA_RS05115: hypothetical protein; CLIBASIA_RS01595: DUF1036 domain-containing protein; CLIBASIA_RS01230: electron transfer flavoprotein subunit alpha/FixB family protein; CLIBASIA_RS01735: DNA gyrase inhibitor YacG; CLIBASIA_RS01125: riboflavin synthase; CLIBASIA_RS03885: PilZ domain-containing protein; CLIBASIA_RS04735: DUF59 domain-containing

**Table 1 | Number of annotated and uncharacterized proteins in the CLas_whole and CLas_HC networks**

| Network | Annotated | Uncharacterized | Total |
|---|---|---|---|
| CLas_whole | 371 | 171 | 542 |
| CLas_HC | 295 | 160 | 455 |

**Table 2 | Interactions between annotated and uncharacterized proteins**

| Interaction type | CLas_whole Node | CLas_whole edge | CLas_HC node | CLas_HC edge |
|---|---|---|---|---|
| Annotated - annotated | 339 | 1491 | 246 | 692 |
| Annotated - hypothetical | 502 | 1976 | 388 | 1106 |
| Hypothetical - hypothetical | 148 | 570 | 123 | 388 |
| Interologs (annotated – annotated) | 128 | 171 | 96 | 106 |
| Interologs (annotated – hypothetical) | 47 | 30 | 30 | 18 |
| Interologs (hypothetical - hypothetical) | 12 | 7 | 7 | 4 |
| Total | 542 | 4245 | 455 | 2314 |

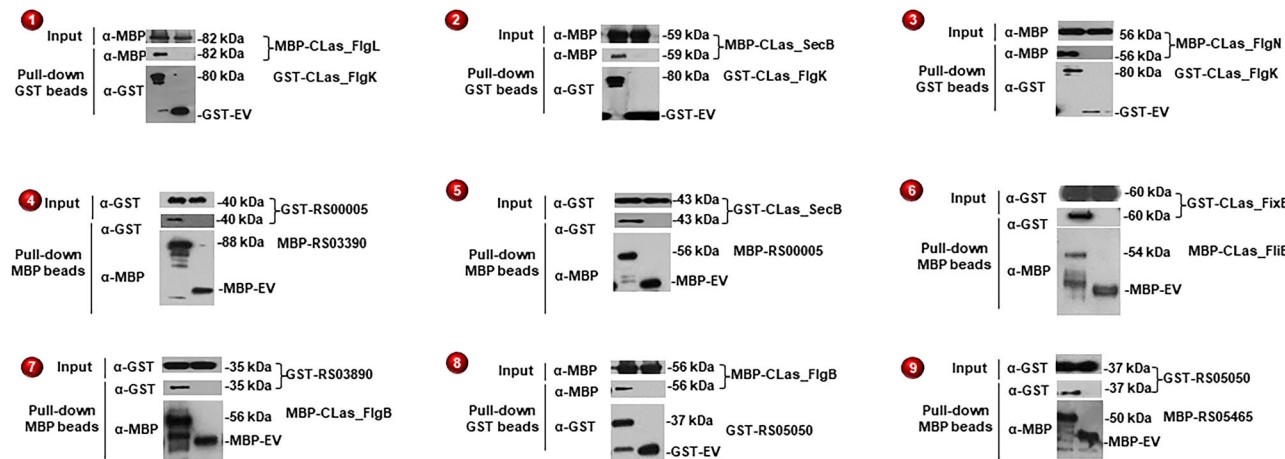

**Fig. 2 | Confirmation of the PPIs identified in Y2H using pull-down assays.** Both GST and MBP pull-down assays were conducted, dependent on protein pairs. Interactions confirmed in vitro are numbered for illustration in Fig. 5. EV- empty vector. CLIBASIA_ was removed from the CLas protein access ID in 4, 5, 7, 8, and 9. Each experiment was repeated independently at least twice with similar results. Source data are provided as a Source data file.

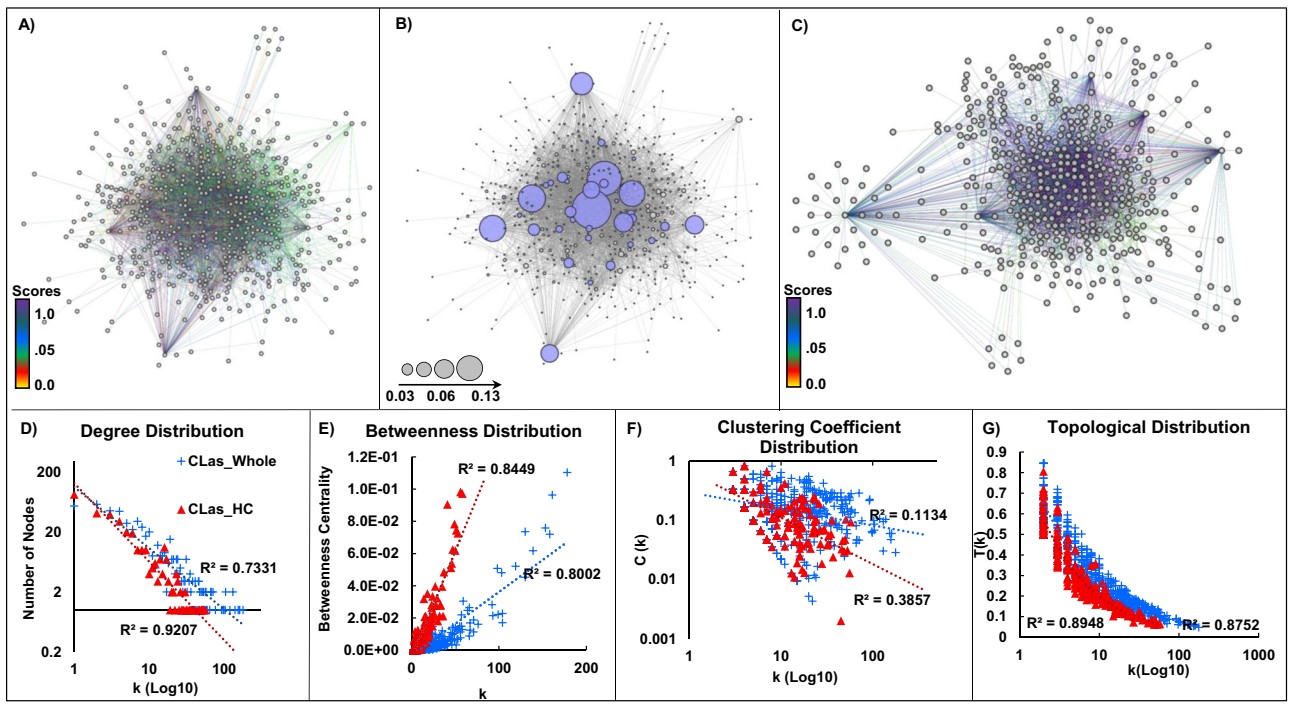

**Fig. 3 | The CLas Y2H network topology. A–C** Graphic representation. n node indicating a protein in the network, e edge indicating the connection between the interacting proteins. **A** CLas_whole network (542n,4245e). **B** hub nodes in the CLas_whole network are colored blue; node size is proportional to its betweenness centrality value. **C** CLas_HC network; PPI confidence scores ≥0.5. **D–G** Network topology of the CLas_whole and CLas_HC networks. **D** Degree distribution. **E** Betweenness centrality distribution. **F** Clustering coefficient distribution. **G** Topological coefficient distribution.

protein][49], and 24 are known to be essential in other bacteria (Supplementary Data 8)[50].

### Inferring functions of CLas hypothetical proteins

CLas contains 415 uncharacterized proteins, including proteins annotated as hypothetical proteins, unknown function, or not in a COG[5,51]. The CLas_HC network contained 1485 high-confidence interactions involving 160 uncharacterized proteins. The features of these proteins include: 15 have signal peptides that may direct them to the cell membrane or outside the cell; 13 are secreted by a nonclassical pathway that does not involve signal peptides as

previously described by Du et al. [52]; 27 have transmembrane domains that span the cell membrane; and 107 have no assigned COG that indicates their function[53]. One hundred forty-nine uncharacterized proteins interacted with at least one annotated protein, and 105 interacted with two or more proteins with known functions (Supplementary Data 3).

To infer the functions of those uncharacterized proteins based on their interacting partners, we employed four guilt-by-association (GBA) methods: 1. Binary interactions and iPPIs, 2. Operon associations, 3. Protein family associations, and 4. Meta-interactome analysis (Fig. 4)[54–56].

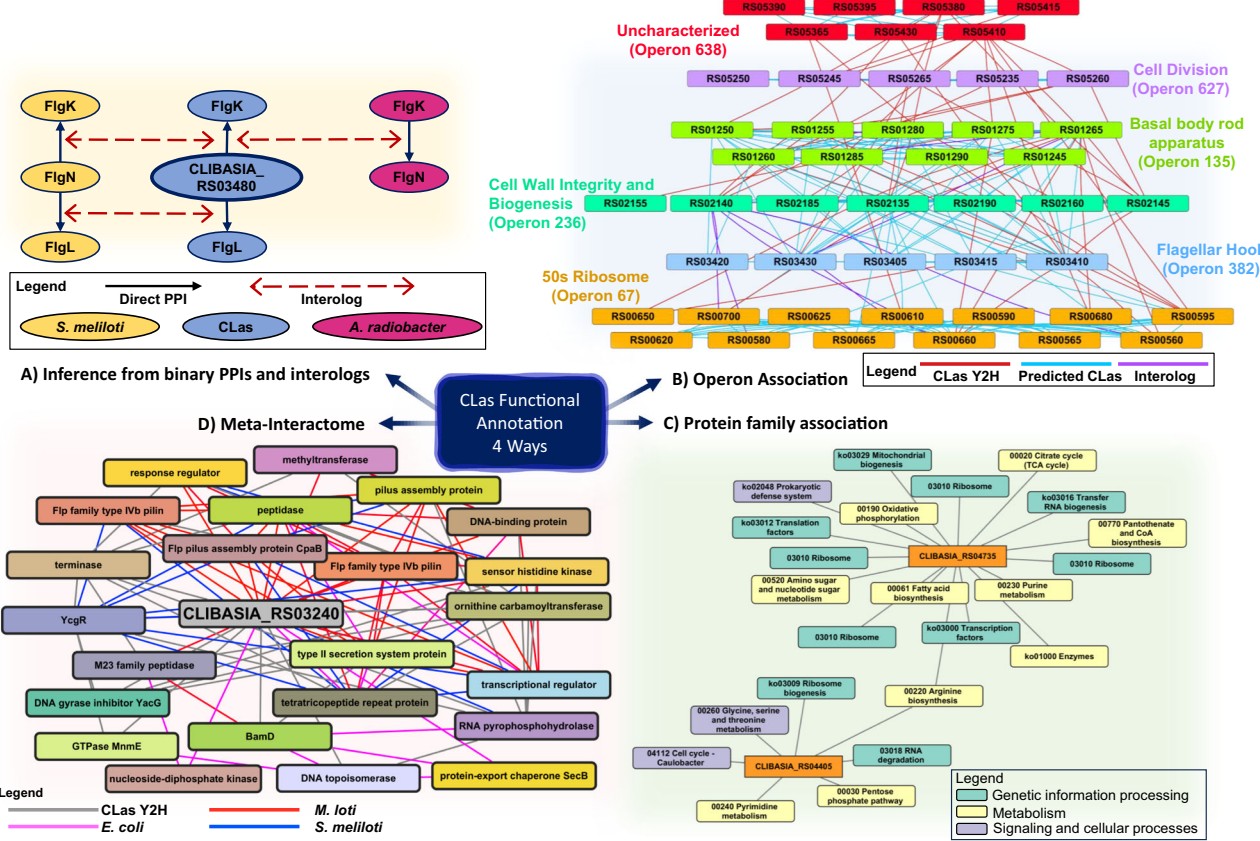

**Fig. 4 | Four approaches were used to infer CLas hypothetical protein functional annotations. A** Hypothetical protein annotations were implied by interologs and binary interactions. These validity of the interactions were confirmed by in vitro assays and tertiary protein structure homology. **B** Inter-operon interactions highlight associations between motility related genes, respiration, membrane integrity, ribosomal proteins, and uncharacterized genes and operons. Operon associations, network and properties are in Supplementary Data 9. CLIBASIA_ was removed from the loci ID for simplicity. **C** Guilt by association was used for CLas proteins interacting with ≥2 proteins in the same protein family or KEGG pathway. CLIBASIA_RS04405 has a functional association with enzymatic proteins, ribosome and RNA biogenesis, and exosome, cytoskeleton proteins. The binary interactions between CLIBASIA_RS04735 and ribosome, transcription/translation proteins implies a functional association with cell response and growth. **D** The sub network meta-interactome mY2H₃₂₄₀ for CLIBASIA_RS03240 inferred ortholog associations with protein export, secretion, kinase, pilus, and peptidase proteins.

## Inferring CLas protein functions from binary interactions

iPPIs are conserved interactions between proteins whose homologs interact in other organisms[13]. As aforementioned, we identified 635 iPPIs between 208 protein pairs in the CLas_whole network (Supplementary Data 5 and 6). Among them, 34 were uncharacterized proteins. We have assigned functions to four proteins based on interacting partners and similarities between tertiary (3D) protein structures (Supplementary Fig. 4). CLIBASIA_RS03480 interacted with FlgK_Las and FlgL_Las in the interactome; the interactions between these proteins are well documented and are iPPIs found in *S. meliloti* and *A. radiobacter* (Fig. 4A)[57]. The protein structure alignment of CLIBASIA_RS03480 in Phyre2 found it shares 31% structure similarity (confidence score of 83.2) with the *Bradyrhizobium sp*. FlgN protein (Uniprot: A0A1I3GD01 (https://www.uniprot.org/uniprotkb/A0A1I3GD01/entry)) whereas they share 17.89% sequence identity. CLIBASIA_RS03480 was reannotated as FlgN_Las, an FlgK and FlgL specific flagellar switch chaperone because it interacted with FlgK_Las and FlgL_Las proteins, involved in flagellar hook formation in the CLas network and its structure homology with the *Bradyrhizobium sp*. FlgN protein structure template (Fig. 4A, Supplementary Fig. 4A). Similarly, we reannotated CLISA-SIA_RS03395 as FliK_Las, a flagellar hook-length regulator, because of its interactions with FlgD_Las, FlhB_Las, and FliP_Las (Supplementary Fig. 4B), three proteins involved in flagellar motor assembly[58–60]. In 2020, MotD from *S. meliloti* was characterized as a flagellar-hook-length regulator and reassigned as FliK; MotD in other α-proteobacteria was reassigned

as FliK dependent on an average AA sequence identity of ≥45% in its C-terminus. CLIBASIA_RS03395, a CLas protein annotated as a chemotaxis protein, shares iPPIs in *S. meliloti*, and the C-terminus protein structure alignment has a 68.83% identity and confidence score of 90.2 compared to the *S. meliloti* FliK protein (Uniprot: F7X8H0 (https://www.uniprot.org/uniprotkb/F7X8H0/entry)). Additionally, CLas hypothetical proteins CLIBASIA_RS03285 and CLIBASIA_RS03885 were annotated as TolA_Las and YcgR_Las respectively (Supplementary Fig. 4C, D). CLIBASIA_RS03285 interacts with TolB in the CLas interactome. The interactions between these two proteins, TolA and TolB, are required for a functional Tol-Pal system in bacteria to maintain cell membrane integrity. Using Phyre2, we found that CLIBASIA_RS03285 has a 45.63% protein structure identity and a confidence score of 100 when aligned with the *E. coli* TolA (Uniprot: P19934 (https://www.uniprot.org/uniprotkb/P19934/entry)) C-terminal domain albeit they only share 23.36% sequence identity. CLIBASIA_RS03885, a PilZ domain-containing protein, is likely a YcgR flagellar brake protein. The protein structure alignment between CLIBASIA_RS03885 and *E. coli* YcgR (Uniprot: P76010 (https://www.uniprot.org/uniprotkb/P76010/entry)) has 45.63% identity and confidence score of 98, while they share 25.93% sequence identity.

## Inter- and Intra- operon binary interaction topology

Next, we took advantage of the fact that bacterial genes of related functions are transcribed together under a single promoter as

operons[61,62]. We reasoned that operons with known proteins interacting with the operons containing uncharacterized proteins provide functional associations/clues for those groups of uncharacterized proteins. For this purpose, we only considered inter- and intra-operon interactions where two or more proteins from each operon interacted together for further analysis (Supplementary Data 9). We examined how the PPIs of other organisms relate to the CLas Y2H network, specifically inter- and intra-operon PPIs (Fig. 4B and Supplementary Fig. 5). We found that CLas had more inter-operon PPIs than intra-operon interactions (Supplementary Data 9 and 10) compared to interactions between homologous proteins. These inter-operon interactions provided evidence of functional association of three CLas operons (operon 638, operon 603, and operon 643), which consist of 21 uncharacterized proteins. Specifically, our interactome data demonstrated that CLas operon 638 has a functional association with flagellar assembly, stress response-related proteins, 50 s ribosomal proteins, cell division, and cell wall integrity proteins (Fig. 4B). In addition, CLas operon 603 has a functional association with translation, FeS cluster assembly, and succinate dehydrogenase. CLas operon 643 has a functional association with metabolism, enzymes, and translation; like operon 603, operon 643 also shares an association with FeS cluster assembly and succinate dehydrogenase (Supplementary Fig. 5).

### Protein family associations

We evaluated PPIs between hypothetical proteins and CLas proteins within known protein families or functions from the Kyoto Encyclopedia of Genes and Genomes (KEGG)[63]. Sixty-one CLas proteins with unknown functions interacted with two or more proteins from the same protein family or function in the CLas_HC Y2H network. These interactors were primarily involved in metabolism and genetic information processing, as shown in examples of protein association networks for CLIBASIA_RS04735 and CLIBASIA_RS04405, which primarily interacted with translation, ribosomal, and enzyme proteins, indicating their functional relationship with metabolism and genetic information processing proteins (Fig. 4C). Operon associations for eleven GBA reannotated CLas proteins confirmed the KEGG protein associations, increasing confidence in their functional assignment. (Supplementary Data 11, 12; Table 2, Supplementary Table 2).

### CLas meta-interactome

As a final sweep of the Y2H data, we looked for associations in a meta-interactome[16]. We generated a meta-interactome (mY2H) using experimentally confirmed PPIs for *A. radiobacter, B. subtilis, C. jejuni, E. coli, H. pylori, L. crescens, M. genitalium, M. loti, M. pneumoniae, S. cerevisiae, S. meliloti,* and *T. pallidum.* The CLas mY2H has 12,780 edges between its 542 proteins; 416 CLas proteins have orthologs with organisms with available interactome maps (Fig. 4D, Supplementary Fig. 6). *M. loti* is more closely related to CLas than other organisms with available Y2H interactome maps, such as *E. coli* and *H. pylori*[9,21,22]. Consequently, most interactions (not including the CLas_whole Y2H PPIs) in the mY2H were PPIs in *M. loti,* totaling 2296 edges. Of these mY2H PPIs, 635 iPPIs between 208 protein pairs were conserved in the CLas_whole, whereas 4037 were not (Supplementary Data 13). We enriched the CLas_whole by adding first neighbor proteins from the ortholog networks for CLas uncharacterized proteins. We infer the functions of 26 uncharacterized CLas proteins based on their interactions with known proteins (Supplementary Fig. 6; Supplementary Data 12; Table 2, Supplementary Table 2). For example, CLIBASIA_RS03240 (Fig. 4D) is a hypothetical protein interacting with the outer membrane protein BamD, DNA topoisomerases, and a GTPase protein in the CLas Y2H. Its meta-interactome was enriched for interactions among protein export, secretion, kinase, pilus, and peptidase proteins. We found CLIBASIA_RS03240 has a high-confidence structure similarity (score of 100) and 59% structure alignment, with 43%

sequence identity to the *Bacillus halodurans* [TaxId:86665] PDB 2nly, a divergent polysaccharide deacetylase (Pfam PF04748), which is consistent with its recent reannotation in the NCBI database. Similarly, CLIBASIA_RS02190, a DUF1009 domain-containing protein, has a high-confidence structure similarity to the lipid A biosynthetic pathway protein LpxI (confidence score of 100, 98% structure alignment, and 31% sequence alignment) from *Caulobacter vibrioides*, and associates with lipid biosynthesis and cell growth-related proteins in the Y2H interactome. Lastly, CLIBASIA_RS01535 is a hypothetical protein that has a high-confidence structure similarity to the GcrA cell cycle regulatory protein (5Z7I) from *Caulobacter crescentus* with a confidence of 100, 96% structure alignment, and 64% sequence identity. It has also been reannotated as such in the NCBI database.

### Novel interactions identified for CLas proteins with known functions

The CLas_whole network identified 4245 interactions between 542 proteins (Fig. 1B). Among the 371 CLas proteins with known functions (Tables 1, 2), we identified 1654 interactions, whereas their homologs had 171 unique experimentally verified interactions in previous studies[9,16–19,21–24,64]. This study identified 1483 novel PPIs that have not been previously reported for proteins with known functions (Supplementary Data 14).

### Y2H PPIs related to flagellar proteins

Flagellar motility might play critical roles in CLas infection of different organs of psyllids and movement in phloem tissues. Importantly, active movement of CLas was observed in the phloem tissue against the flow of phloem sap[65,66]. We paid close attention to flagella-related proteins. Our Y2H screening detected 399 interactions among known flagellar proteins in our Y2H screening (Fig. 5; Supplementary Data 15; Supplementary Tables 3, 4). In addition, nine interactions were confirmed with pull-down assays (Fig. 2). Of those PPIs confirmed, two were putative secreted proteins, CLIBASIA_RS05050[53] and CLIBASIA_RS05465[52], interacting with each other and FlgB$_{Las}$, a flagellar basal-body protein[67]. Another secreted protein, CLIBASIA_RS03890, also interacted with FlgB$_{Las}$. These interactions imply that these proteins may play a role in flagellar formation and functions, which could affect the pathogenicity or survival of CLas[68]. FlgN is a known flagellar export chaperone of FlgK and FlgL, two hook-filament junction proteins[57]. We also identified SecB, a chaperone protein dedicated to translocating proteins across the cytoplasmic membrane, as an interactor of FlgK. We validated both interactions, FlgK$_{Las}$-FlgN$_{Las}$ and FlgK$_{Las}$-SecB$_{Las}$, consistent with the involvement of the Sec pathway in flagellar synthesis[69]. Moreover, SecB$_{Las}$ interacted with CLIBASIA_RS0005, a protein with a T-SNARE-like domain. T-SNARE proteins mediate membrane fusion in eukaryotic cells. Pathogens can exploit host SNARE proteins to enter host cells by displaying SNARE-like domains on their surface membrane[69]. It remains to be determined whether CLIBASIA_RS0005 is involved in CLas entering cells or glands of psyllids[70].

## Discussion

We have constructed a large-scale protein-protein interaction map of CLas using a high-throughput Y2H approach. In total, we have identified 4245 PPIs among CLas proteins. We have employed a three-phase screening approach to improve the robustness of the Y2H. We estimated that our high-throughput Y2H data has a false positive rate of 2.9%, consistent with previously reported false positive rates of 0.5%[38], less than 5%[14], and 6.5%[71]. The pull-down assay supported the robustness of our Y2H with the verification of nine PPIs found in the Y2H network. Among the 1662 interactions between proteins with known functions of CLas, only 171 have been observed in other systems, indicating that CLas proteins might have evolved new functions to adapt to the reduced genome size. CLas, *T. pallidum*, and

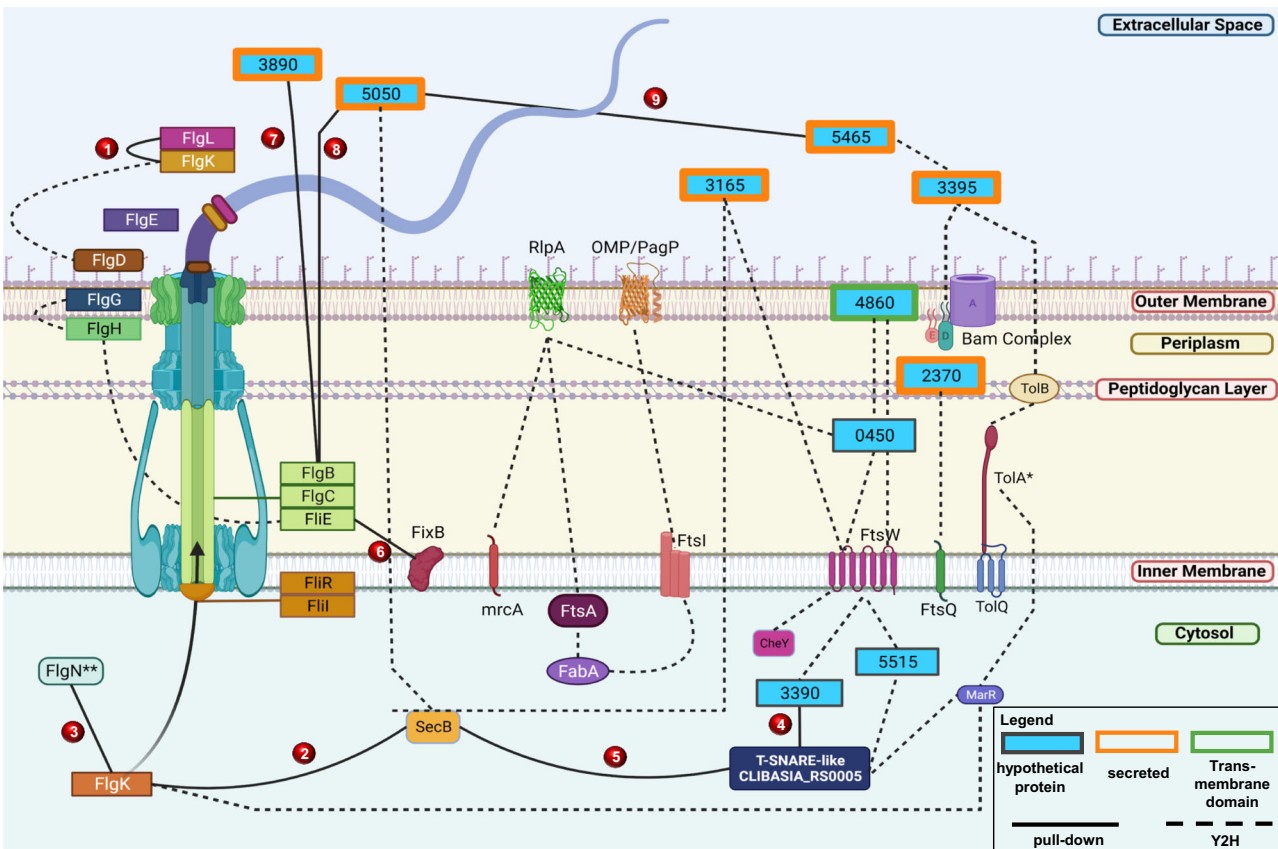

**Fig. 5 | Integrated view of CLas proteins related to flagellar biosynthesis and membrane proteins, their interacting partners identified in Y2H.** Subcellular localization representation of CLas hypothetical and annotated protein interactions. The source of the interactions and hypothetical proteins are denoted by dashed and solid lines. Hypothetical proteins with orange borders are experimentally confirmed secreted proteins as described in Prasad et al. and Du et al. Interactions confirmed with pull-down experiments were numbered from 1–9 in red circles. The hypothetical proteins were named using their CLIBASIA_RS#. **Here CLIBASIA_RS03840 is represented as FlgN. This figure was generated using BioRender.

*M. pneumoniae* have a greater number of interactions per node in their PPI networks than free-living bacteria with larger genomes, such as *E. coli, M. loti, H. pylori, C. jejuni,* and *B. subtilis.* An increase in node neighbors seems to be a common phenomenon for bacteria that have undergone reductive evolution[45].

We identified 40 putative hub nodes in the network[47]. Hub proteins might act as moonlighting proteins and have multiple functions as suggested by their multiple interacting partners[72]. It remains to be determined whether intrinsically disordered proteins (IDPs) and regions (IDRs) are involved in the multitasking of moonlighting proteins[73]. Hub proteins are suggested to be essential to the organism or involved in critical PPIs[23,41,74]. On the contrary, such findings were partially debunked in a study by Yu et al.[75]. Three hundred fourteen genes are required for *L. crescens* growth in pure culture, and the CLas genome encodes 238 homologs to these *L. crescens* genes[49]. However, CLas is missing 76 essential *L. crescens* genes, suggesting that CLas has different, and possibly more, physiological needs for growth than *L. crescens.* Hub proteins may provide insight into culturing CLas since we may be able to identify critical proteins necessary for its growth in culture. Eight of the hub proteins were reported to be required for culturing *L. crescens*[49], and 24 are known to be essential in other bacteria[50].

Furthermore, ten of the 40 hub genes are significantly up- or down-regulated in either citrus or the ACP vector, suggesting they play crucial roles for CLas survival in the host or vector[76]. This differential expression could allude to lifestyle switches in the bacterium during the host-vector transition and may be exploited for antibiotic or other targeted strategies for HLB management. Several CLas hub proteins have antioxidant capabilities, protecting the bacteria from oxidative stress damage, including quinone oxidoreductase 1[77] and LysR-type transcriptional regulator[78]. Quinone oxidoreductase 1 acts as a cytoplasmic antioxidant, whereas LysR is a key circuit component in regulating microbial stress responses and is required for bacterial tolerance to $H_2O_2$ in vitro[78]. The protective mechanism of CLas is consistent with the model that HLB is a pathogen-triggered immune disease; CLas triggers systemic and chronic production of reactive oxygen species (ROS), leading to phloem tissue cell death and HLB symptoms[79–81]. Thus, those proteins will protect CLas from the high ROS levels in the phloem tissues. Another hub protein, CtrA, regulates bacterial cells, such as *Caulobacter crescentus,* to generate cells of different morphologies, with and without flagella[82]. CLas was also reported to be polymorphic in shape[40]. CtrA is a master cell cycle regulator, negatively regulating the septum-inhibiting proteins, altering the location FtsZ from the middle of the cell to a polar end of the replicating cell. This alteration causes the cell to split without symmetry, producing elongated polymorphic cells and what are known as mini cells. CtrA may be involved in the non-flagellated and polymorphic phenotypes of CLas cells in phloem tissues. Nineteen of the 40 hub proteins are uncharacterized, requiring further investigation into their role in CLas biology. However, some hub nodes were given functional associations; for example, CLIBASIA_RS00015 associates with metabolism, DNA replication, and signaling proteins, and CLIBASIA_RS05475 associates with metabolism, enzymes, translation, FeS cluster assembly, and succinate dehydrogenase proteins; these associations were determined from the KEGG protein family guilt by association and operon association analysis. For example, Phyre2

predicted that CLIBASIA_RS00015 is an ATP-dependent deoxyribonuclease subunit. The prediction for CLIBASIA_RS00015 is consistent with the interactome network functional associations assigned as a metabolism protein interacting with DNA replication since ATP-dependent deoxyribonuclease subunits play a role in DNA recombination.

Two hundred and eight binary PPIs, of the 4245 PPIs detected in this study, had been reported in other organisms[8,9,16–23,64]. On the other hand, 4037 novel interactions were identified in this study, expanding the PPI database. We identified two thousand five hundred eighty-three interactions involving 171 uncharacterized CLas proteins, providing clues for their functions. For instance, we have assigned protein functions to CLIBASIA_RS03480 as FlgN$_{Las}$, CLIBASIA_RS03395 as FliK$_{Las}$, CLIBASIA_RS03285 as TolA$_{Las}$, and CLIBASIA_RS03885 as YcgR$_{Las}$. CLIBASIA_RS04860 and CLIBASIA_RS02370 might play a role in cell division since they interact with FtsW$_{Las}$ and FtsQ$_{Las}$, two essential components of the divisome complex[83]. These proteins interact with a hub protein CLIBASIA_RS00450, CLIBASIA_RS05515, and RlpA (septal ring lytic transglycosylase, CLIBASIA_RS04130). RlpA is known to be involved in cell separation and rod shape of bacteria[84]. Thus, even though the functions of CLIBASIA_RS04860, CLIBASIA_RS02370, CLIBASIA_RS00450, and CLIBASIA_RS05515 are unknown, they are probably involved in cell division or cell wall formation, providing hints for further characterization.

We have identified 37 uncharacterized proteins that interact with multiple proteins of an operon, and 61 uncharacterized proteins interacted with two or more proteins from the same KEGG family or function, providing solid indications regarding their roles associated with the known functions of the operons or protein families compared to single interactions. Interestingly, CLas seems to encode most flagellar genes even though flagella have not been observed in CLas *in planta*[40]. Though flagella have been observed for CLas in the psyllid vector, supported by recent plant vs. psyllid transcriptome data showing higher levels of flagellar gene expression in the psyllid versus expression in the plant, The extensive PPIs involving flagellar proteins suggest they are functional and consistent with the active movement of CLas against the flow of phloem sap[66].

Overall, we have generated the interactome map for CLas, which has provided insights regarding the biology and pathogenicity of CLas and the putative functions of uncharacterized proteins. This interactome will be a valuable resource and provide clues to further characterize unknown proteins' functions. The PPIs identified have the potential to be used as targets for the development of novel antimicrobials to control HLB.

## Methods

### Yeast two-hybrid vector construction
Full-length CLas ORFs were PCR amplified using genomic DNA extracted from CLas-positive citrus leaf tissues. CLas protein information including Uniprot IDs and annotations are included in Supplementary Data 16. The gel-purified amplicons were fused with the activation (AD) and binding domain (BD) Y2H expression vectors (pGADT7 and pGEO_BD, respectively) using infusion recombinase cloning (Clontech). We modified the Gal4 Matchmaker Gold Yeast Two-hybrid system (clonetech) pGBK_T7 (Clontech) expression vector to harbor the pGADT7_AD vector's multiple cloning site (MCS) between RE sites NdeI and BamHI (Supplementary Fig. 1B) and was named pGEO_BD. This modification simplified the cloning of all CLas ORFs into both vectors between RE sites EcoRI and XmaI using one infusion primer set per gene to reduce costs and labor. We directly transformed these AD and BD infusion reactions (~20 μl per 50 μl reaction) into yeast strains Y187 (AD, prey clones; MATa; G1::lacZ, M1::MEL1) and Y2H Gold (BD, bait clones; MATα; G1::HIS3, G2::ADE2, M1::AUR1-C, M1::MEL1; G1, G2, and M1 are Gal4 driven promoters)

following Yeastmaker Yeast Transformation protocol (Takara). We verified transformants with colony PCR and sequencing a random set of ~200 constructs.

### Autoactivation test
Autoactivation of the reporter genes by individual ORFs was assessed by mating the transformants with the opposite corresponding mating partner empty vector and plating on selective media double dropout (DDO)/X/AbA (-Leu, -Trp, +XαGal, +Aureobasidin A). Clones that autonomously activated reporter genes were removed from the 96-well plates and high throughput screening.

### Yeast two-hybrid three-phase screening
A three-phase 96-well mating scheme was adapted from a matrix approach[33]. Briefly, all transformants were cultured individually in 96-well plates in 1 ml of the appropriate minimal liquid media (AD: -Leu; BD: -Trp) at 30 °C until saturated (~36–48 h). For the first mating phase (three phases in total), all individual 1 ml AD cultures were pooled together and centrifuged at 700 g for 5 min. The pellet was resuspended in 25 ml SD/-Leu liquid media with 25% glycerol. Aliquots (1 ml) of the pool were stored at −80 °C. The BD cultures were left in a 96-well format for the phase 1 mating.

### Mating procedure
Phase 1. Each AD pool was mated to individual pGEO_BD clones in a 96-well format. The matings were spotted on YPDA agar plates by spotting three μl of the AD pool aliquot and three μl of the individual BD clone on top of one another and incubated at 30 °C for ~48 h, or until the colonies were ~1 mm in diameter. The 1 mm colonies were transferred using a pin replicator to 96-well plates with DDO (-Leu, -Trp) liquid medium to select diploids harboring AD and BD plasmids and incubated at 30 °C and 210 rpm on a rotary shaker. This step reduces the background on the selection plates. After 48 h, 5 μl of the mating cultures were plated on DDO (-Leu, -Trp) agar plates as a mating control, and selective media to determine interaction and interaction intensity: D/X/A (-Leu, -Trp, +XαGal, +Aureobasidin A), Q/X (-Leu, -Trp, -His, -Ade + XαGal), and Q/X/A (-Leu, -Trp, -His, -Ade +XαGal, +Aureobasidin A). All subsequent matings and screenings were carried out in the same manner. Phase 2. Positive BD clones from the phase 1 mating were pooled using the previously described AD pooling method. These BD pools (containing all positive BD interactors from Phase 1) were mated back to the AD plate that comprised the pool of Phase 1. This is to determine which of the AD clones within the pool were the interacting partners to the positive BD interactors from phase 1 (Fig. 1A). Phase 3. The final phase determined the interacting partners from the previous two screenings and allowed for a pairwise rescreening of all interacting constructs. All positive AD and BD yeast constructs interacting in the first two phase screenings were organized into a row (BD) and column (AD) format in 96-well plates. Each row of a 96-well plate is a single BD construct (For example, row A: BD construct #1, row B: BD construct #2, row C: BD construct #3... to row H: BD construct #8) and each column is a single AD construct (For example column 1: AD construct #1, column 2: AD construct #2, column 3: AD construct #3... to column 12: AD construct #12) which were mated pairwise. The row and column format allowed all phase one and two proteins to interact to determine the interacting partners and the interaction specificity. This method helped determine what interactions are more likely or accurate since interactions must be repeatable pairwise and allowed us to weed out sticky interactors.

### Analysis of the CLas Y2H network topology
We visualized, analyzed, and utilized the network developed in this study with Cytoscape network software[43]. Network topologies, such as betweenness and closeness centrality, degree, and clustering

coefficient, were determined using the Network Analyzer in Cytoscape[44]. We used this information to select hub nodes in the network for further analysis and to understand general CLas interactome network organization found in the Y2H screening. To validate the CLas_whole network topology, NetworkRandomizer version 3 was used to compare the CLas_whole network to a randomly computed network. We compared the difference between the real network (CLas_whole) and its most similar random network for each centrality[85]. Hub nodes were determined using manual curation, MCODE clustering, and MCC algorithm[46,86,87].

## Scoring interactions and assigning interaction confidence values

PPIs were scored on a scale from one to three for each selection media. These colony scores were used with iPPIs, PPIs within the same operon, and bait out-degree and prey in-degree to create an additive score for the logistic regression formula to assign interaction confidence scores. The logistic regression training set of 100 iPPIs was used as true positives and 100 PPIs with the lowest combined Y2H score and highest average shortest path link value with randomized bait and prey node degree values as true negatives. We calculated the interaction score ($s$) using the Eq. (1) four times using different variables for each regression. The variables were the number of times as a bait ($b$), number of times as a prey ($p$), and colony score ($c$). We calculated the interaction scores for each PPI by averaging four interaction scores, which were calculated independently based on the value of the independent variable ($c$), i.e., interolog score alone, Y2H repeatability, interaction intensity, and interolog score combined with Y2H repeatability together. We compared the scores within each group of interactions and found that they were all within one standard deviation of their group mean[88]. This average score is termed interaction score in the datasets.

$$s = 1/(1 + e^{-(-178.67 - (-2.25484)*b - 0.4624*p + 506.1149*c)}) \tag{1}$$

$s$: interaction score; $b$: count of occurrence as a bait node; $p$: count of event as a prey node; $c$: colony score.

## Conserved protein-protein interactions (iPPIs)

CLas homologs in *A. radiobacter, B. subtilis, C. jejuni, E. coli, H. pylori, L. crescens, M. genitalium, M. loti, M. pneumoniae, S. cerevisiae, S. meliloti, T. pallidum*, and iCOGs were collected using IMG/MER and NCBI databases[17,22,36,88,89]. Only protein sequences with an identity of >30% were considered for analysis. All available experimentally determined (i.e., Y2H, AP/MS, and CO-IP) ortholog PPIs and iCOG PPIs were collected from the STRING database for these homologs[36]. The known and predicted interactions between CLas proteins were also included in the data collected from STRING[36].

## Validating CLas interactions in vivo by a pairwise Y2H screening

We randomly chose 163 ORF constructs (15.9% of the total CLas ORFs and 30% of the nodes in the CLas_whole network) to use in a pairwise Y2H screening. The interactions from this screening were compared for overlap between the three datasets: the pairwise screening, the three-phase Y2H PPI screening, and the known and predicted interactions between these proteins available from the STRING database. The rates of false positives, false negatives, specificity, and sensitivity within the three-phase screening were determined using the known interactions and pairwise screening. The false positive rate (FPR) was calculated by comparing the number of PPIs detected in the high throughput screening and not found in the pairwise rescreening using the FPR = FP/(FP + TP). The false positive (FP) number is the total PPIs found in the CLas_whole high throughput dataset not found in the STRING database as predicted for the random 163 proteins screened

pairwise. The true positive (TP) value represents the number of protein pairs that did not interact in either screening method. True negative (TN) rates (TN=n(n-1)/2; $n$ = PPIs found only in the high throughput screening) are difficult to determine because there is no definitive method for proving whether proteins do or do not interact in vivo. The TN value was calculated using predicted/iPPIs by comparing the high throughput results with the STRING expected PPIs and the 163 random protein pairs rescreened pairwise to increase the likelihood of finding true interactors for reference. The false negative rate (FNR) was calculated similarly (FNR = FN/(TP + FN)), where the false negative (FN) value is the number of PPIs detected pairwise but not in the high throughput screening.

The true positive (TP) value is the number of PPIs found in the high throughput and pairwise screenings. The FNR for the CLas interactome shows the ratio of PPIs expected but not found in the CLas_whole. Specificity and sensitivity were calculated using the above TP, TN, FN values and the false positive (FP) number, which is the number of PPIs detected in the pairwise but not in the high throughput dataset. Sensitivity is the proportion of true PPIs detected of all protein pairs that interact in the known dataset, represented by the formula: Sensitivity = TP/(TP + FN). The specificity represents the proportion of non-interactors correctly excluded from the dataset, represented by the formula: Specificity = TN/(FP + TN).

## Confirming CLas interactions using a pull-down assay

Interactions were validated using MBP/GST pull-down assay. The pGEX-4T-1 (GST) (GE Healthcare) and pMAL-c5x (MBP) (NEB) vectors were used to construct the *E. coli* expression vectors for GST-Glutathione or MBP-Amylose pull-down assay between multiple CLas proteins. The GST/MBP expression vectors were constructed by amplifying the ORFs from the CLas yeast expression vectors and inserted between RE sites: BamHI and SalI in the pGEX vector and XmnI and EcoRI in the pMAL vector using infusion cloning (Clontech). These newly generated vector constructs were transformed into *E. coli* Rosetta (DE3) electrocompetent cells, transformants were selected by plating on LB agar plates containing ampicillin (100 μg/ml) and chloramphenicol (30 μg/ml); all subsequent steps used these antibiotics at these concentrations. The transformants were cultured overnight in 3 ml liquid LB. The next day, the culture was diluted 1:100 in 25 ml LB and grown until the culture reached an $OD_{600}$ 0.2–0.3. The culture was induced with IPTG (1 μg/ml) overnight at 16 °C.

After induction, the resulting cell pellet was lysed using 5 μg/ml of pellet with B-PER™ Bacterial Protein Extraction Reagent (Thermo-Fisher) and incubated at room temperature for 20 min while rocking. The lysates were cleared by centrifugation at 4 °C for 20 min at 16,000 g. Supernatants were collected in a fresh tube, and total protein concentration was measured using Bradford assay. The protein expression and solubility were evaluated by both Coomassie stain and western blot (WB) of the GST and MBP fusion proteins using anti-GST (ab92, Abcam, 1:1000 dilution) and anti-MBP (E8032S, NEB, 1:10,000 dilution) monoclonal antibodies before proceeding. The GST-fusion protein and empty-GST supernatants were mixed with the MBP-fusion protein and incubated with glutathione or amylose agarose beads for 3–4 h at 4 °C with rotating. The glutathione or amylose agarose was then washed 5–10 times to remove unbound proteins before boiling and analysis by SDS-PAGE WB using anti-MBP (NEB) and anti-GST (Abcam) monoclonal antibodies followed by secondary antibody Goat Anti-Mouse IgG H&L (HRP) (ab205719, Abcam, 1:10,000). Protein-protein interactions were further confirmed by pull-down assays in Supplementary Table 5.

## Reporting summary

Further information on research design is available in the Nature Portfolio Reporting Summary linked to this article.

## Data availability
The datasets generated and/or analysed during the current study are available in the Supplementary Information, Supplementary Data and Source Data files. Source data are provided with this paper.

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

## Acknowledgements

We thank Wang lab members for constructive suggestions and insightful discussions. This project was supported by funding from Florida Citrus Initiative Program, Citrus Research and Development Foundation, U.S. Department of Agriculture National Institute of Food and Agriculture grants 2022-70029-38471, 2021-67013-34588, 2018-70016-27412 and 2016-70016-24833, FDACS Specialty Crop Block Grant Program, and Hatch project [FLA-CRC-005979] (N. Wang).

## Author contributions

N.W. and E.W.C. conceptualized and designed the experiments. E.W.C. and O.G.P. designed and constructed the pGEO_BD vector and oligonucleotides. E.W.C. performed the experiments. E.W.C. and N.W. wrote the manuscript with input from all co-authors.

## Competing interests

The authors declare no competing interests.
