## [Peer Review File · Nature Communications]

The protein interactome of the citrus Huanglongbing pathogen *Candidatus Liberibacter asiaticus*Reviewer #1 (Remarks to the Author):

The paper by Carter et al. is an extraordinary tour de force to characterize the protein interactome of CLAs, the unculturable bacterium associated with citrus greening disease of citrus using yeast 2 hybrid. Their study is extraordinarily impactful, highlighting new biology about the pathogen and its ability to survive in its host with a reduced genome. Their discovery of 1491 novel protein interactions between CLAs proteins with known (or rather previously described functions in other organisms) clearly demonstrates that our knowledge of bacterial protein interactions is still extremely limited and restricted to a few well studied species that are culturable and/or medically significant. Using interlog analysis, they assigned functions to previously unknown functions based on their interacting partners and 3D structures. Notably, these proteins are involved in flagellar biology. This is significant because there has been debate in the field on whether CLAs forms functional flagella (Barnett et al. 2019 PNAS), a key component for pathogenesis across a range of bacterial species. This point can be better highlighted in their discussion. The methods used are sound and the work meets the standards expected in the field. There is enough detail in the methods for the work to be reproduced. The authors have gone above and beyond to provide supplementary data and associated analyses.

There are minor grammatical issues in the introduction – mainly pertaining to tenses of some sentences (the authors switching from present tense to past tense unnecessarily). This could be cleaned up a bit before publication.

Line 60 – Introduction. The authors describe studies using Y2H to study interactions in other organisms but leave out viruses. I suggest including reference to a couple of references describing host-virus interactomes by Y2H.

Line 61 - The authors mention AP-MS detects interactions in the native organism, which is true. But they further mention that AP-MS is thus not ideally suited for non-cultured microorganisms. However, this is not true. AP-MS has been used extensively to study the interactomes of plant and animal-infecting viruses.

Line 132 - The authors show a very interesting trend whereby the CLAs network has many node neighbors, consistent with the idea that protein complexity increases with genome reduction. The authors may want to also consider in their hypothesis that the intracellular lifestyle and need to function within a plant or insect cell may also drive protein multifunctionality in CLAs. This raises the question in my mind as to whether there is also an increase in protein disorder in the proteins at the HUBs or network nodes with many other protein interactors. If the authors have done a disorder analysis, it would be a great addition to include and discuss in the context of these findings. Additionally, the authors may want to mention and cite Henderson and Martin 2011 (Infection and Immunity) in the context of discussing these data. Highly conserved proteins involved in metabolic regulation or cell stress response can have a range of other biological functions including in bacterial virulence.

Line 147 – The authors report that eight of the HUB proteins are shared with *L. crescens* and 24 are known to be essential in other bacteria. It is not easy to find what HUBs the authors are referring to in this sentence specifically. Table DS8 seems to contain most of these data but it is not entirely clear in the table which of the 40 are referred to in that sentence. Moreover, the reference to the De Francesco et al. 2022 paper in DS8 is confusing – what is the fold change column referring to? More information is needed about those data and the relevance to the data in the table.

Reviewer #2 (Remarks to the Author):

see attached file.

Reviewer #2 Attachment on the following page

Report on Carter et al., The protein interactome of [...] Candidatus Liberibacter asiaticus

Carter et al. describe a new protein-protein interaction map of Liberibacter, an alpha-proteobacterium for which (I believe) no other such dataset exists (besides the data in their preprint). Hence, the study is welcome new information to a poorly studied bacterial proteome and is thus certainly worth reporting.

The work seems to be solid, and overall, the methodology looks sound and reproducible -- with all caveats for this kind of analysis (see below for details).

That said, there are a number of issues that can and should be addressed.

First, I recommend to add a more precise strain name to the paper, and ideally a (NCBI) taxon ID, or a RefSeq ID (apparently NC_012985.3), given that multiple strains of CLAs have been sequenced. Ideally, authors should also add a reference number for a proteome in Uniprot, which produces 27 proteomes when I search the Uniprot proteomes for the taxonID associated with the protein IDs provided in Data set DS1. (Carter et al. say that their genome encodes 1,027 proteins, but none of the Uniprot CLAs proteomes has exactly that number).

Methods

(see also below; this section only refers to methods mentioned in the Introduction or Results).

Line 95. Authors should clarify if the the PPIs from STRING are only physical interactions or if they include ALL (functional) interactions from STRING, including co-expressed proteins.

Line 105. The False Positive and False Negative rates (3.1% and 47.9%) sound very optimistic – and I doubt they are realistic. That’s also indicated by the average node degree of 15 and 10 in both whole and HC networks as well as Figure 1c, in which only a small fraction of the STRING PPIs is found (and Extended Data Table S1 which gives node degrees for other interactomes).

Results

Line 140. The claim that highly connected proteins are more likely to be essential has been partially debunked and has a couple of caveats, e.g. by Yu et al. 2008, Science 322: 104ff, 10.1126/science.1158684.

Line 153-154. How do you know that “13 are secreted by a nonclassical pathway”? I suspect these are predicted or have they been experimentally determined to be secreted?

Line 168. If CLIBASIA_RS03480 was reannotated as FlgN, there should be a comment on sequence similarity and structural similarity (as shown in Extended Data Figure S4).

Line 171. Authors should provide a few more details on their reannotation of “CLISASIA_RS03395 as FliK”; they mention a similarity to the FliK structure only in Extended Data Figures S4B, but not in the text (they should; also whether there is any sequence similarity). It may be helpful to mention any operon data, co-expression, or phylogenetic profile in the Extended Data Figure.

Line 174. Similar for CLIBASIA_RS03285 and CLIBASIA_RS03885 which are only cursorily mentioned in the text.

Discussion

Line 254. Are these 208 PPIs listed in any table? They should.

Line 260 says that “CLIBASIA_RS04960 and CLIBASIA_RS02370 might play a role in cell division since they interact with FtsW_{Las} and FtsQ_{Las}, but these interactions are not shown in Figure 5.

If these and other proteins are involved in cell division, there is likely a phenotype in some homolog that has been deleted or mutated. Authors should double-check in some phenotyping studies that measure cell division and comment on that (cell division proteins are rarely essential but often show a visible phenotype, such as elongated cells).

Line 290. It is indeed interesting that “CLas seems to encode most flagellar genes even though flagella have not been observed in CLas”. It may be worthwhile to produce a table with flagellar proteins in CLas and some flagellated (and unflagellated) relatives, so see if there is any correlation with the absence or presence of certain proteins.

Methods

Authors may want to consider to deposit their ORF clones in a public repository, such as BEI Resources.

Is the vector pGEO_BD described and published elsewhere? If not, please provide details and ideally cite a sequence. Also, authors may want to consider to deposit pGEO_BD in a public repository, such as AddGene.

Lines 312 ff. Re: yeast strains Y187 (AD, prey clones; MAT_a; G1::lacZ, M1::MEL1) and Y2H Gold (BD, bait clones; MAT_α; G1::HIS3, G2::ADE2, M1::AUR1-C, M1::MEL1). May want to mention that G1, G2, and M1 are promoters and that Clontech is now Takara.

Line 318. I suspect DDO stands for double dropout, but not sure, so please spell out.

Lines 380ff. References given don't match IMG/NCBI sources for PPIs, esp. ref. 16. See comments on Datasets below.

Line 383. Ref 16 is not about STRING.

Lines 391 ff. The description of how FNs and FPs were determined are too cursory and do not explain how this was done.

Line 419. T ?

References

More references are incomplete, only giving a volume number without further details. Only some have DOIs.

Figures

Figure 1. Insert (high confidence) after HC

Figure 2. This needs more labels to be clearer. I suggest to add molecular masses to the input rows (e.g. on the right-hand side of the pluses/minuses). Also, I assume EV stands for empty vector, but this needs to be spelled out. Sample 6 says both CLas and CLAS.

Figure 4a. The CLas proteins are very dark and the text is barely visible when printed.

Figure 4b. The individual genes in the operons should be numbered by their actual locus numbers. All having the same number makes no sense. Besides, the legend makes no sense, as it says "*B. Hypothetical protein annotations were implied by interologs and binary interactions. These validity of the interactions were confirmed by in vitro assays and tertiary protein structure homology*".

Figure 4c. Please add legends for box colors (what do they mean?). For the enzymes, what are they? What are the numbers?

Figure 4d. CLIBASIA_RS05095 doesn't seem to be a good example, as it is nowhere discussed in the text (at least I didn't find it), and the inferred function is pretty vague. Based on the interactions shown it's involved in either Translation, DNA replication, mRNA degradation (misspelled as degredation) or metabolism, so this is not very informative. I am sure the authors can find better examples in their dataset.

Figure 5. PPI 6 does not match PPI 6 in Fig. 2. What are the asterisks on FlgN?

Tables

I suggest to move Extended Data Tables S2 and S3 to the main text.

Extended Data Table S7: please include common names, if available; the systematic names are not very informative. Also, it's not very practical to have this in the pdf, may be better in dataset.

Supplementary data / Datasets / spreadsheet

This file needs a table of contents sheet, explaining all sheets.
I would also freeze panes.

DS1. I would add a reference / URL where the protein IDs can be found. I do find the proteins in Uniprot, but when I search for the first one, WP_012778342.1, Uniprot has 2 proteins under this ID, so this is confusing (I guess both entries have the same sequence, but I haven't checked). Maybe add Uniprot IDs.

Regarding the taxon ID in Uniprot, CLIBASIA_RS00005 is listed as belonging to strain psy62 which has 1,103 entries in Uniprot, so that doesn't match the number given in the introduction.

DS2. I wonder what the interaction score means, especially when it is = 1, which indicates a perfect score.

DS3. "Ineractome" should be interactome.

DS4. I don't understand what COG means here, given that COG doesn't have Y2H etc. data. Please explain.

DS5. "teh"; explain column R, notes. How were these interologs mapped to STRING?

DS6. I don't recall what the iCOGs in STRING refer to. There should be a source interactome, so what is it?

DS12. Spell out SDE; Notes: numbers appear to indicate operons, but need to explain (or simply spell out as Operon 638 etc.)

DS14. This is somewhat redundant, just saying ...

Reviewer #3 (Remarks to the Author):

The authors present a Y2H-based map of a bacterial citrus pathogen that is well motivated due to the difficulties in culturing this bacteria resulting in it being difficult to study. They use their interaction map to annotate genes of previously unknown function. The experimental methods appear to be sound. I think there are several major issues with the analysis that should be addressed.

Major issues

- Fig 1C - the number of PPIs found in the pairwise test but not in the 3-step pooling (71 PPIs) is very similar to the number from the pooling but not the pairwise test (64 PPIs). So I disagree with the statement L108 "probably resulting from the pooling". I think the data indicates that the sampling from the pooling is close to saturation relative to a pairwise test and that the false negative rate is an intrinsic limitation of the Y2H assay, which is only able to detect a fraction of PPIs and/or proteins (the same is true for other binary PPI assays, see Venkatesan et al. Nature Methods 2009, Braun et al. Nature Methods 2009, Choi et al. Nature Communications 2019).
- Fig 1C - L102 "Each CLas Y2H screening detected 41 of the 117 predicted interactions among these 163 proteins" This is not what the Venn diagram shows. It shows 41 were detected by both methods, 52 by the pooled approach and 51 by the pairwise approach.
- L104 - there is not enough information in the methods to understand exactly how the 3.1% and 47.9% were arrived at. Please add the details of the calculations.
- I don't see how the false positive rate can be calculated from the data obtained. I guess it is using pairs from the screening that are not recovered in the pairwise test? Ideally for this calculation, the best-practice to calculate the false positive rate is to re-test a random sample of >100 PPIs from the dataset in a different PPI assay and compare to positive and negative reference sets (Venkatesan et al. Nature Methods 2009). Testing some PPIs in pull-downs (Fig. 2) is great, and the data looked good, but it's a small number and wasn't a random sample.
- L386 the authors state that the subspace of 163 ORFs was randomly chosen but the interaction density of that space is much higher than the whole space: $\sim 896 / (163^2 / 2) = 6.7\%$ vs $\sim 4,245 / (916 \times 936 / 2) = 1\%$. Can the authors confirm that those ORFs were randomly chosen from the ~ 900 clones. If they were instead selected from the 542 that gave interactions, that would explain the higher density but would mean that the sensitivity calculations would be wrong.
- Extended Data Fig 3 - I do not understand the bottom panel. What's the difference between Average number of interactions "per ORF" and "per node"? Maybe per ORF is the number in each genome, whereas per node is the number with at least one PPI in each network. Needs to be clearer.
- Extended Data Fig 3 - the idea that organisms with smaller genomes could end up, on average, with proteins of higher degree, is a very interesting one. However I'm not convinced by the evidence presented. Comparing different interactome mapping efforts, performed by different labs at different times is dangerous because any observed difference is most likely a technical artifact. The most obvious way is from the differing completeness of the different interactome maps. One suggestion for an imperfect way to test for the effect of completeness would be to add an analysis that uses equation 8 from Stumpf et al. PNAS 2008 to estimate the complete interactome size for each species using the node incompleteness. Also I wouldn't include the AP-MS dataset in the analysis, and be careful about including the yeast dataset. Also, it would be better to directly plot what you are testing: the genome size vs average degree.
- L365: I'm confused about the confidence score calculation. Why was the regression done four times and averaged, instead of using all the variables together, as it is usually done? What is the rationale for taking the longest shortest path in random networks for the negative set?
- Fig 3D-G - I think the differences in network topology between the full and high-confidence networks can largely be explained by the use of degree in calculating the confidence score. This is not mentioned in the text.
- L146 the essential genes numbers have no context / p-value. The authors should compare the frequency of essential genes in and outside of the hubs with a statistical test. It would not be surprising if they are not enriched since Yu et al. Science 2008 showed that using a systematic Y2H map, degree did not correlate with essentiality but with pleiotropy instead and that the earlier observations of the degree essentiality relationship were mediated through the confounding factors of study bias in literature-curated PPI networks and through abundance in systematic AP/MS.
- L189 "CLas operon 638 has a functional association with flagellar assembly, stress response-

related proteins, 50s ribosomal proteins, cell division, and cell wall integrity proteins". This seems like a lot of different processes for one operon?

Minor issues / suggestions

- Fig. 1D - the binned plot loses a lot of information on the confidence score. Better to plot the data in a continuous way, e.g. the number of PPIs above the threshold, as the score is increased from 0-1.
- I would briefly state which variables are input to the confidence score in the main text.
- It would be helpful to show the confidence score distribution for the interologs in the supplementary figures
- The word hub is not usually in all-caps (HUB).
- For MCC, state the full name "Maximal Clique Centrality (MCC)" to avoid confusion with e.g. Mathews correlation coefficient.
- For MCODE the original reference should be added: An automated method for finding molecular complexes in large protein interaction networks, Bader & Hogue, BMC Bioinformatics (2003).
- Extended data table S1 - Uetz is the last author of the Nature Biotech 2014 E. coli interactome, so should be Rajagopala et al.
- Extended Data Fig. S3 - legend says bacteria but figure includes yeast
- Extended Data Fig. S3 - you use the term genome size twice, to mean two different things, I think on the x-axis it is the number of protein-coding genes and the bubble size is the number of bases
- Maybe clearer to the reader to calculate average degree rather than average number of interactions per node?
- L254: the 208 PPIs already reported were in other species, presumably with homologous but not identical proteins? If that's the case, then you're really the first to report all your PPIs.

Responses to reviewers' comments

Reviewer #1

The paper by Carter et al. is an extraordinary tour de force to characterize the protein interactome of CLas, the unculturable bacterium associated with citrus greening disease of citrus using yeast 2 hybrid. Their study is extraordinarily impactful, highlighting new biology about the pathogen and its ability to survive in its host with a reduced genome. Their discovery of 1491 novel protein interactions between CLas proteins with known (or rather previously described functions in other organisms) clearly demonstrates that our knowledge of bacterial protein interactions is still extremely limited and restricted to a few well studied species that are culturable and/or medically significant. Using interlog analysis, they assigned functions to previously unknown functions based on their interacting partners and 3D structures. Notably, these proteins are involved in flagellar biology. This is significant because there has been debate in the field on whether CLas forms functional flagella (Barnett et al. 2019 PNAS), a key component for pathogenesis across a range of bacterial species. This point can be better highlighted in their discussion. The methods used are sound and the work meets the standards expected in the field. There is enough detail in the methods for the work to be reproduced. The authors have gone above and beyond to provide supplementary data and associated analyses.

Response: We thank the reviewer for the positive comments. It is always encouraging to know our colleague appreciate our work. We have highlighted flagella related in the discussion as suggested.

There are minor grammatical issues in the introduction – mainly pertaining to tenses of some sentences (the authors switching from present tense to past tense unnecessarily). This could be cleaned up a bit before publication.

Response: Thanks for the suggestions. We have changed the tenses. Two sentences with mixed tenses were corrected in the introduction for consistency.

Line 60 – Introduction. The authors describe studies using Y2H to study interactions in other organisms but leave out viruses. I suggest including reference to a couple of references describing host-virus interactomes by Y2H.

Response: We have added the following: “viruses including potyviruses Soybean mosaic virus (PMID: 19189854)” as suggested.

Line 61 - The authors mention AP-MS detects interactions in the native organism, which is true. But they further mention that AP-MS is thus not ideally suited for non-cultured microorganisms. However, this is not true. AP-MS has been used extensively to study the interactomes of plant and animal-infecting viruses.

Response: We thank the reviewer for the excellent suggestion. We have revised it accordingly.

Line 132 - The authors show a very interesting trend whereby the CLas network has many node neighbors, consistent with the idea that protein complexity increases with genome reduction. The authors may want to also consider in their hypothesis that the intracellular lifestyle and need to function within a plant or insect cell may also drive protein multifunctionality in CLas. This raises the question in my mind as to whether there is also an increase in protein disorder in the proteins at the HUBs or network nodes with many other protein interactors. If the authors have done a disorder analysis, it would be a great addition to include and discuss in the context of these findings. Additionally, the authors may want to mention and cite Henderson and Martin 2011 (Infection and Immunity) in the context of discussing these data. Highly conserved proteins involved in metabolic regulation or cell stress response can have a range of other biological functions including in bacterial virulence.

Response: Thanks for the insightful suggestion. After literature review on protein disorder, we decided not to conduct the disorder analysis owing to the low accuracy of known predictors. Necci et al. compared the performance of all disorder predictors. It was reported that “Disordered binding regions remain hard to predict, with $F_{\max} = 0.231$.” (Necci, M., Piovesan, D., CAID Predictors. *et al.* Critical assessment of protein intrinsic disorder prediction. *Nat Methods* **18**, 472–481 (2021).). For PPIs studies, the binding regions are the most critical, but with such a low accuracy, it is unlikely to provide robust information. However, we have taken intrinsically disordered proteins (IDPs) and regions (IDRs) into our discussion as shown below. In addition, we also cited Henderson and Martin 2011 (PMID: 21646455) in the revised manuscript.

“It remains to be determined whether intrinsically disordered proteins (IDPs) and regions (IDRs) are involved in the multitasking of moonlighting proteins (PMID: 33875885).”

Line 147 – The authors report that eight of the HUB proteins are shared with *L. crescens* and 24 are known to be essential in other bacteria. It is not easy to find what HUBs the authors are referring to in this sentence specifically. Table DS8 seems to contain most of these data but it is not entirely clear in the table which of the 40 are referred to in that sentence. Moreover, the reference to the De Francesco et al. 2022 paper in DS8 is confusing – what is the fold change column referring to? More information is needed about those data and the relevance to the data in the table.

Response: We have listed these hubs that are also essential for *L. crescens* growth in culture in the text. We have also added a clearer description of the De Francesco et al. data in the table legend.

Reviewer #2

Carter et al. describe a new protein-protein interaction map of *Liberibacter*, an alpha-proteobacterium for which (I believe) no other such dataset exists (besides the data in their preprint). Hence, the study is welcome new information to a poorly studied bacterial proteome and is thus certainly worth reporting.

The work seems to be solid, and overall, the methodology looks sound and reproducible -- with all caveats for this kind of analysis (see below for details).

Response: We sincerely appreciate the critical but constructive reviews and the positive comments.

First, I recommend to add a more precise strain name to the paper, and ideally a (NCBI) taxon ID, or a RefSeq ID (apparently NC_012985.3), given that multiple strains of CLAs have been sequenced. Ideally, authors should also add a reference number for a proteome in Uniprot, which produces 27 proteomes when I search the Uniprot proteomes for the taxonID associated with the protein IDs provided in Data set DS1. (Carter et al. say that their genome encodes 1,027 proteins, but none of the Uniprot CLAs proteomes has exactly that number).

Response: The RefSeq ID has been added. The manuscript was also updated with more detail pertaining to the gene number. We used the annotation table available in the Duan et al 2009 manuscript of the first sequence of CLAs for these number as reference. This study defined 1136 genes encoded in the genome, 1027 of which are nonredundant and protein coding. 612 are annotated with some function and 415 are uncharacterized (69 have general function, 46 are unknown function, 300 are without a COG family or ID).

Methods

(see also below; this section only refers to methods mentioned in the Introduction or Results).

Line 95. Authors should clarify if the the PPIs from STRING are only physical interactions or if they include ALL (functional) interactions from STRING, including co-expressed proteins.

Response: We used experimentally confirmed PPIs for orthologs which were used as predictive PPIs for CLAs. We stated: However, there were 474 and 369 experimentally determined interactions between homologous, interolog proteins (iPPIs) and interacting protein family interologs (iCOGs) at confidence cutoffs ≥ 0.4 and ≥ 0.9 , respectively (Table 1; Extended Data Table S2).

Line 105. The False Positive and False Negative rates (3.1% and 47.9%) sound very optimistic – and I doubt they are realistic. That’s also indicated by the average node degree of 15 and 10 in both whole and HC networks as well as Figure 1c, in which only a small fraction of the STRING PPIs is found (and Extended Data Table S1 which gives node degrees for other interactomes).

Response: We have clarified the method in the methods section “Validating CLAs interactions *in vivo* by a pairwise Y2H screening” for calculating the false positive and negative rates and have rechecked our calculations independently. The calculations were conducted using the high

throughput dataset and random sampling of proteins and pairwise rescreening, these results were compared to the expected PPIs found in other organisms predicted for CLAs. Additionally, the false positive rates of Y2H were previously reported to be 0.5%⁷³, less than 5%⁷⁴, and 6.5%⁷⁵. Our false positive rate is consistent with previous studies. We think our low false positive rate is at the expense of high false negative rate. The false negative rate for our high throughput study may explain why there is only a small fraction of STRING PPIs found. Our high throughput data was highly reproducible between randomly sampled proteins subjected to a pairwise screening.

Results

Line 140. The claim that highly connected proteins are more likely to be essential has been partially debunked and has a couple of caveats, e.g. by Yu et al. 2008, Science 322: 104ff, 10.1126/science.1158684.

Response: Thanks for the insightful comment. We have added the following to take this study into consideration: "On the contrary, such findings were partially debunked in a study by Yu et al. (PMID: 18719252)."

Line 153-154. How do you know that "13 are secreted by a nonclassical pathway"? I suspect these are predicted or have they been experimentally determined to be secreted?

Response: These non-classically secreted proteins were experimentally determined using an Escherichia coli alkaline phosphatase assay by Du, P. *et al.* 2021.

Line 168. If CLIBASIA_RS03480 was reannotated as FlgN, there should be a comment on sequence similarity and structural similarity (as shown in Extended Data Figure S4).

Response: revised as suggested.

Line 171. Authors should provide a few more details on their reannotation of "CLISASIA_RS03395 as FliK"; they mention a similarity to the FliK structure only in Extended Data Figures S4B, but not in the text (they should; also whether there is any sequence similarity). It may be helpful to mention any operon data, co-expression, or phylogenetic profile in the Extended Data Figure.

Response: revised as suggested.

Line 174. Similar for CLIBASIA_RS03285 and CLIBASIA_RS03885 which are only cursorily mentioned in the text.

Response: revised as suggested.

Discussion

Line 254. Are these 208 PPIs listed in any table? They should.

Response: Yes, the CLas locus IDs and the homolog IDs are listed in datasets 5 & 6.

Line 260 says that “CLIBASIA_RS04960 and CLIBASIA_RS02370 might play a role in cell division since they interact with FtsWLas and FtsQLas, but these interactions are not shown in Figure 5.

Response: The PPI between 4860 and FtsW has been added to figure 5, the PPI between 2370 and FtsQ is already shown in the figure.

If these and other proteins are involved in cell division, there is likely a phenotype in some homolog that has been deleted or mutated. Authors should double-check in some phenotyping studies that measure cell division and comment on that (cell division proteins are rarely essential but often show a visible phenotype, such as elongated cells).

Response: Because CLas has not been cultivated, no phenotypes are available for the mutants of the genes mentioned above.

Line 290. It is indeed interesting that “CLas seems to encode most flagellar genes even though flagella have not been observed in CLas”. It may be worthwhile to produce a table with flagellar proteins in CLas and some flagellated (and unflagellated) relatives, so see if there is any correlation with the absence or presence of certain proteins.

Response: Similar work comparing the conservation of flagellar genes across other Rhizobiales was done in our previous study: Andrade, M. O., Pang, Z., Achor, D. S., Wang, H., Yao, T., Singer, B. H., & Wang, N. The flagella of ‘Candidatus Liberibacter asiaticus’ and its movement in planta. *Molecular plant pathology*. 21(1), 109-123 (2020).

Methods

Authors may want to consider to deposit their ORF clones in a public repository, such as BEI Resources.

Response: We will look into BEI since we had not used it before. Meanwhile, the ORF clones will be available for interested parties based on Material Transfer Agreement which is also a commonly used approach.

Is the vector pGEO_BD described and published elsewhere? If not, please provide details and ideally cite a sequence. Also, authors may want to consider depositing pGEO_BD in a public repository, such as AddGene.

Response: The vector pGEO_BD has not been published before and has been described in this study with details in term of how it was constructed. This vector will be deposited into Addgene.

Lines 312 ff. Re: yeast strains Y187 (AD, prey clones; MAT α ; G1::lacZ, M1::MEL1) and Y2H Gold (BD, bait clones; MAT α ; G1::HIS3, G2::ADE2, M1::AUR1-C, M1::MEL1). May want to mention that G1, G2, and M1 are promoters and that Clontech is now Takara.

Response: The promoters are now mentioned in the methods and Clontech to Takara is also updated.

Line 318. I suspect DDO stands for double dropout, but not sure, so please spell out. Lines 380ff. References given don't match IMG/NCBI sources for PPIs, esp. ref. 16. See comments on Datasets below.

Response: References have been fixed and DDO has been spelled out with the first abbreviation.

Line 383. Ref 16 is not about STRING.

Response: revised.

Lines 391 ff. The description of how FNs and FPs were determined are too cursory and do not explain how this was done.

Response: This has been updated so to explain in detail in the methods section: "Validating CLas interactions *in vivo* by a pairwise Y2H screening"

Line 419. T ?

Response: Typo, removed

References

More references are incomplete, only giving a volume number without further details. Only some have DOIs.

Response: Fixed

Figures

Figure 1. Insert (high confidence) after HC

Response: Fixed

Figure 2. This needs more labels to be clearer. I suggest to add molecular masses to the input rows (e.g. on the right-hand side of the pluses/minuses). Also, I assume EV stands for empty vector, but this needs to be spelled out. Sample 6 says both CLas and CLAS.

Response: Thank you for the suggestion, this has been updated and fixed in the figure 2.

Figure 4a. The CLas proteins are very dark and the text is barely visible when printed.

Response: Fixed

Figure 4b. The individual genes in the operons should be numbered by their actual locus numbers. All having the same number makes no sense.

Response: Revised as suggested.

Besides, the legend makes no sense, as it says “B. Hypothetical protein annotations were implied by interologs and binary interactions. These validity of the interactions were confirmed by in vitro assays and tertiary protein structure homology”.

Response: The order of the figure was changed “B” should go with “A” and vice versa, this caption is an oversight and has been corrected.

Figure 4c. Please add legends for box colors (what do they mean?). For the enzymes, what are they? What are the numbers?

Response: The legend has been added and we cleaned up the figure to make it more clear/readable. We have also replaced the general KEGG annotation with the most detailed available. The numbers represented the number of CLAs proteins within that particular KEGG category, these have now been removed since they were superfluous.

Figure 4d. CLIBASIA_RS05095 doesn’t seem to be a good example, as it is nowhere discussed in the text (at least I didn’t find it), and the inferred function is pretty vague. Based on the interactions shown it’s involved in either Translation, DNA replication, mRNA degradation (misspelled as degredation) or metabolism, so this is not very informative. I am sure the authors can find better examples in their dataset.

Response: We found multiple better examples such as CLIBASIA_RS03240 for this method and have updated the figure. Also, this example has been included in the text in both the Meta-interactome results and the discussion.

Figure 5. PPI 6 does not match PPI 6 in Fig. 2. What are the asterisks on FlgN?

Response: Fixed the PPI for #6 and added information to FlgN** in the figure caption.

Tables

I suggest to move Extended Data Tables S2 and S3 to the main text.

Response: Revised as suggested.

Extended Data Table S7: please include common names, if available; the systematic names are not very informative. Also, it’s not very practical to have this in the pdf, may be better in dataset.

Response: Common names have been added in the table (the new Extended Data Table S4).

Supplementary data / Datasets / spreadsheet

This file needs a table of contents sheet, explaining all sheets.

Response: Revised as suggested.

I would also freeze panes. DS1. I would add a reference / URL where the protein IDs can be found. I do find the proteins in Uniprot, but when I search for the first one, WP_012778342.1, Uniprot has 2 proteins under this ID, so this is confusing (I guess both entries have the same sequence, but I haven’t checked). Maybe add Uniprot IDs.

Response: Revised as suggested.

Regarding the taxon ID in Uniprot, CLIBASIA_RS00005 is listed as belonging to strain psy62 which has 1,103 entries in Uniprot, so that doesn't match the number given in the introduction.

Response: Strain name, seq reference, and gene count have been fixed and clarified.

DS2. I wonder what the interaction score means, especially when it is = 1, which indicates a perfect score.

Response: This score was determined using interaction intensity, number of times a protein acted as either bait or prey, and whether the protein is interacting as an interolog (conserved protein). Yes, a 1 is considered a perfect score. An explanation has been added to the dataset caption.

DS3. "Ineractome" should be interactome.

Response: Fixed

DS4. I don't understand what COG means here, given that COG doesn't have Y2H etc. data.

Response: The String database now lets users search for PPIs that have been reported in organisms between proteins belonging to a COG. This allows for a more thorough query for PPIs when not much evidence is available. In our case, there is limited Y2H bacterial interactome data and experimentally determined data for CLas. Using the COG String PPI tool, we were able to expand our conserved PPI dataset found in our Y2H study to organisms outside of our specific searches, which included closely related and model organisms.

DS5. "teh"; explain column R, notes. How were these interologs mapped to STRING?

Response: Dataset 6 lists the homolog locus ids for the interologs in dataset 5. Dataset 5 is in attempt to show clearly the repeated interologs across queried species and COG PPIs.

DS6. I don't recall what the iCOGs in STRING refer to. There should be a source interactome, so what is it?

Response: As mentioned earlier, String gives COG association-PPI data. For our study, we used the term iCOG to represent COG association-PPI data that we found as interologs in the CLas Y2H network.

DS12. Spell out SDE; Notes: numbers appear to indicate operons, but need to explain (or simply spell out as Operon 638 etc.)

Response: Fixed SDE in the table caption to be spelled out and also used your suggestion for spelling our Operon and #.

DS14. This is somewhat redundant, just saying ...

Response: Because of the complexity of the data, we kept this one to help readers to get the data more easily.

Reviewer #3

The authors present a Y2H-based map of a bacterial citrus pathogen that is well motivated due to the difficulties in culturing this bacteria resulting in it being difficult to study. They use their interaction map to annotate genes of previously unknown function. The experimental methods appear to be sound. I think there are several major issues with the analysis that should be addressed.

Response: We sincerely appreciate the positive comments and also the critical but constructive suggestions.

Major issues

- Fig 1C - the number of PPIs found in the pairwise test but not in the 3-step pooling (71 PPIs) is very similar to the number from the pooling but not the pairwise test (64 PPIs). So I disagree with the statement L108 "probably resulting from the pooling". I think the data indicates that the sampling from the pooling is close to saturation relative to a pairwise test and that the false negative rate is an intrinsic limitation of the Y2H assay, which is only able to detect a fraction of PPIs and/or proteins (the same is true for other binary PPI assays, see Venkatesan et al. Nature Methods 2009, Braun et al. Nature Methods 2009, Choi et al. Nature Communications 2019).

Response: Excellent point. We have revised as follows: Because of the highly reproducible PPIs (92%) between the three-phase and pairwise screening (Figure 1C), it is probable that the false negative rate is an intrinsic limitation of the Y2H assay, which is only able to detect a fraction of PPIs, as previously reported for other binary PPI assays (Venkatesan et al. Nature Methods 2009, Braun et al. Nature Methods 2009, Choi et al. Nature Communications 2019).

- Fig 1C - L102 "Each CLas Y2H screening detected 41 of the 117 predicted interactions among these 163 proteins" This is not what the Venn diagram shows. It shows 41 were detected by both methods, 52 by the pooled approach and 51 by the pairwise approach.

Response: We have revised the sentence as follows: The three-phase screening detected 52 whereas the pairwise approach identified 51 of the 117 predicted interactions among these 163 proteins with 41 detected by both methods (Figure 1C, Supplementary Dataset S4).

- L104 - there is not enough information in the methods to understand exactly how the 3.1% and 47.9% were arrived at. Please add the details of the calculations.

Response: This has been elaborated further in the methods.

- I don't see how the false positive rate can be calculated from the data obtained. I guess it is using pairs from the screening that are not recovered in the pairwise test? Ideally for this calculation, the best-practice to calculate the false positive rate is to re-test a random sample of >100 PPIs from the dataset in a different PPI assay and compare to positive and negative

reference sets (Venkatesan et al. Nature Methods 2009). Testing some PPIs in pull-downs (Fig. 2) is great, and the data looked good, but it's a small number and wasn't a random sample.

Response: We calculated the false positive rates based on the expected and found interactions in addition to the interaction confirmation using a random sampling and alternative method to the three-phase pooling.

Recalculation of Rates:

1. False Positive rate (FPR) = $FP/(FP+TN)$

The false positive rate for the CLas_whole network is 0.029 or 2.9%. Out of all the protein pairs not interacting in the interolog dataset, 2.9% test positive in the CLas_whole network.

2. False negative rate (FNR) = $FN/(TP+FN)$

Of all the possible known interactions (117), 55.6% of them tested negative in the CLas_whole network.

3. Sensitivity = $TP/(TP+FN)$

Proportion of true PPIs detected. Of all the protein pairs that interact in the known dataset, 44.4% test positive in the CLas_whole network.

4. Specificity = $TN/(FP+TN)$

Proportion of non-interactors correctly excluded from the dataset. Of all the protein pairs that do not interact in the known dataset, 97.1% test negative in the CLas_whole network.

- L386 the authors state that the subspace of 163 ORFs was randomly chosen but the interaction density of that space is much higher than the whole space: $\sim 896 / (163^2 / 2) = 6.7\%$ vs $\sim 4,245 / (916 \times 936 / 2) = 1\%$. Can the authors confirm that those ORFs were randomly chosen from the ~ 900 clones. If they were instead selected from the 542 that gave interactions, that would explain the higher density but would mean that the sensitivity calculations would be wrong.

Response: Indeed, these proteins were chosen randomly from all available clones in Yeast expression vectors and not only from interacting constructs. We calculated sensitivity based on the proportion of PPIs detected in the pairwise screening compared to the pooled screening.

- Extended Data Fig 3 - I do not understand the bottom panel. What's the difference between Average number of interactions "per ORF" and "per node"? Maybe per ORF is the number in each genome, whereas per node is the number with at least one PPI in each network. Needs to be clearer.

Response: We agree with your concerns below and have removed Fig. S3.

- Extended Data Fig 3 - the idea that organisms with smaller genomes could end up, on average, with proteins of higher degree, is a very interesting one. However I'm not convinced by the evidence presented.

Response: Kelkar and Ochman 2013 published a study showing how genome reduction of obligate pathogenic and endosymbiotic bacteria causes an increase in protein functional complexity. They hypothesize that with the loss of genes from genome reduction brought on by genetic drift and asexual reproduction (ex: population bottlenecks in the case of insect vector bacteria), the remaining genes will encode proteins who interact with a greater functional range of proteins to compensate for genes lost. They clearly showed a relationship between genome size and the increase in average number of interacting partners with genome reduction using PPIs between 168 broadly conserved proteins; the authors used seven bacteria for their comparison. We are not claiming to be the pioneers for this idea, we are simply comparing the CLas data, and have confirmed Kelker and Ochman's hypothesis stands true in the case of CLas.

Comparing different interactome mapping efforts, performed by different labs at different times is dangerous because any observed difference is most likely a technical artifact. The most obvious way is from the differing completeness of the different interactome maps. One suggestion for an imperfect way to test for the effect of completeness would be to add an analysis that uses equation 8 from Stumpf et al. PNAS 2008 to estimate the complete interactome size for each species using the node incompleteness. Also I wouldn't include the AP-MS dataset in the analysis, and be careful about including the yeast dataset. Also, it would be better to directly plot what you are testing: the genome size vs average degree.

Response: We agree with your concerns and have removed Fig. S3.

- L365: I'm confused about the confidence score calculation. Why was the regression done four times and averaged, instead of using all the variables together, as it is usually done? What is the rationale for taking the longest shortest path in random networks for the negative set?

Response: The regression was calculated using three variables: 1. Number of times as a prey, 2. Number of times as bait, and 3. Colony score (interaction intensity score, repeatability, and interolog score). We performed the regression analysis four times switching out the colony score variable with each calculation and averaging the four regressions, which all were below 1 standard deviation. The PPIs with the longest shortest path link was used as the most likely negative set since these interactions will contribute the least to the communication of the network. This is part of a method for interaction score regression from the *T. pallidum* interactome published by Titz et al 2008 (PMID: 18509523).

- Fig 3D-G - I think the differences in network topology between the full and high-confidence networks can largely be explained by the use of degree in calculating the confidence score. This is not mentioned in the text.

Response: Degree was not the only variable used in determining the high confidence, and not all nodes of high degree were eliminated in the HC network. We disagree that the topology changes are largely to the degree. However, we do agree that the clustering coefficient distribution showed an HC network with more defined subnetworks and a more efficient communication between nodes (represented by the betweenness), which can give the impression that the difference is due to the degree of the nodes. You can see that the distributions for the degree and topological coefficient do not change much between the two datasets (whole and HC). So, yes degree plays into these distributions and topologies, but not the only player.

- L146 the essential genes numbers have no context / p-value. The authors should compare the frequency of essential genes in and outside of the hubs with a statistical test. It would not be surprising if they are not enriched since Yu et al. Science 2008 showed that using a systematic Y2H map, degree did not correlate with essentiality but with pleiotropy instead and that the earlier observations of the degree essentiality relationship were mediated through the confounding factors of study bias in literature-curated PPI networks and through abundance in systematic AP/MS.

Response: It is difficult to know exactly which genes are essential for CLas because it can not be cultured. We can only surmise based on essential gene homologs. This is why we used only communicatively overrepresented nodes in the CLas network to determine putative essential genes. These putative essential genes were then queried against the database of essential genes and the *L. crescens* essential genes for culturing study (Lai et al. 2016) to support our putative nodes. To take your suggestions into consideration, we have revised our manuscripts as follows: "Hub proteins are suggested to be essential to the organism or involved in critical PPIs^{22,38,79,80,82}. On the contrary, such findings were partially debunked in a study by Yu et al. (PMID: 18719252)". This way, we take a neutral position on whether there is a relationship between hub proteins and essentiality.

- L189 "CLas operon 638 has a functional association with flagellar assembly, stress response-related proteins, 50s ribosomal proteins, cell division, and cell wall integrity proteins". This seems like a lot of different processes for one operon?

Response: Yes! Reduced genomes have protein functional complexity. Proteins in these organisms take on multiple roles to compensate for gene loss.

Minor issues / suggestions

- Fig. 1D - the binned plot loses a lot of information on the confidence score. Better to plot the data in a continuous way, e.g. the number of PPIs above the threshold, as the score is increased from 0-1.

Response: We have generated Extended Data Figure S3 as suggested.

- I would briefly state which variables are input to the confidence score in the main text.

Response: We have added more details for determining the interaction confidence scores in the text.

- It would be helpful to show the confidence score distribution for the interologs in the supplementary figures

Response: We have included the confidence score distribution for all PPIs in Extended Data Figure S3.

- The word hub is not usually in all-caps (HUB).

Response: Fixed

- For MCC, state the full name "Maximal Clique Centrality (MCC)" to avoid confusion with e.g. Mathews correlation coefficient.

Response: Fixed

- For MCODE the original reference should be added: An automated method for finding molecular complexes in large protein interaction networks, Bader & Hogue, BMC Bioinformatics (2003).

Response: Fixed

- Extended data table S1 - Uetz is the last author of the Nature Biotech 2014 E. coli interactome, so should be Rajagopala et al.

Response: Fixed

- Extended Data Fig. S3 - legend says bacteria but figure includes yeast

Response: Extended Data Fig. S3 was deleted per your previous comments.

- Extended Data Fig. S3 - you use the term genome size twice, to mean two different things, I think on the x-axis it is the number of protein-coding genes and the bubble size is the number of bases.

Response: Extended Data Fig. S3 was deleted per your previous comments.

- Maybe clearer to the reader to calculate average degree rather than average number of interactions per node?

Response: Extended Data Fig. S3 was deleted per your previous comments.

- L254: the 208 PPIs already reported were in other species, presumably with homologous but not identical proteins? If that's the case, then you're really the first to report all your PPIs.

Response: Yes, this is true. In fact, all of the PPIs here have not been reported for CLAs. However, we felt that saying these PPIs are already reported to give credit for the interolog PPIs found in other studies.

Reviewer #2 (Remarks to the Author):

It seems that most or all of my concerns have been addressed.

One issue I noticed was that there is no explicit section on Results.

I also tend to replace the "Main" section with an Introduction. It's just easier for authors to navigate the paper with clearly demarcated sections and subsections (rather too many than too few).

Reviewer #3 (Remarks to the Author):

Thank you to the authors for their responses. However, a couple of things are still not clear.

It's still not clear to me in the text where the false positive rate 2.9% number comes from. Please just spell out what exactly the numerator and denominator numbers are. If they are numbers from the Venn diagram in Fig 1C, state that, using the same terms in the figure.

Additionally, the authors didn't offer an explanation for why the random subspace has seven times higher density than the overall space. Possibly just that some hub proteins fell into that space, but it would be good to address the question.

REVIEWERS' COMMENTS

Reviewer #2 (Remarks to the Author):

It seems that most or all of my concerns have been addressed.

One issue I noticed was that there is no explicit section on Results.

I also tend to replace the "Main" section with an Introduction. It's just easier for authors to navigate the paper with clearly demarcated sections and subsections (rather too many than too few).

Response: We thank the reviewer for the positive comments and we have replaced “Main” with Introduction, Results, and Discussion.

Reviewer #3 (Remarks to the Author):

Thank you to the authors for their responses. However, a couple of things are still not clear.

It's still not clear to me in the text where the false positive rate 2.9% number comes from. Please just spell out what exactly the numerator and denominator numbers are. If they are numbers from the Venn diagram in Fig 1C, state that, using the same terms in the figure.

Response: Thank you for your comment, the rates were, in fact, determined using the Venn diagram values. We have updated the manuscript so that this is clear.

Additionally, the authors didn't offer an explanation for why the random subspace has seven times higher density than the overall space. Possibly just that some hub proteins fell into that space, but it would be good to address the question.

Response: Yes, the randomly selected proteins for the independent screening did inevitably include ten hub nodes in the random selection. Here they are listed:

no.	protein ID	CLas ID
1	WP_012778733.1	CLIBASIA_RS01985
2	WP_015452358.1	CLIBASIA_RS01545
3	WP_015452511.1	CLIBASIA_RS02780
4	WP_015452552.1	CLIBASIA_RS02995
5	WP_015452634.1	CLIBASIA_RS03410
6	WP_015453015.1	CLIBASIA_RS05300

7	WP_012778599.1	CLIBASIA_RS01285
8	WP_015452715.1	CLIBASIA_RS03885
9	WP_015452527.1	CLIBASIA_RS02865
10	WP_012778566.1	CLIBASIA_RS01125

This explains the density difference between the high throughput dataset and the random subset screening.